# Symport and antiport mechanisms of human glutamate transporters

Biao Qiu [1] ✉ & Olga Boudker [1,2] ✉

Excitatory amino acid transporters (EAATs) uptake glutamate into glial cells and neurons. EAATs achieve million-fold transmitter gradients by symporting it with three sodium ions and a proton, and countertransporting a potassium ion via an elevator mechanism. Despite the availability of structures, the symport and antiport mechanisms still need to be clarified. We report high-resolution cryo-EM structures of human EAAT3 bound to the neurotransmitter glutamate with symported ions, potassium ions, sodium ions alone, or without ligands. We show that an evolutionarily conserved occluded translocation intermediate has a dramatically higher affinity for the neurotransmitter and the countertransported potassium ion than outward- or inward-facing transporters and plays a crucial role in ion coupling. We propose a comprehensive ion coupling mechanism involving a choreographed interplay between bound solutes, conformations of conserved amino acid motifs, and movements of the gating hairpin and the substrate-binding domain.

Human excitatory amino acid transporters (EAATs) concentrate the neurotransmitter glutamate and aspartate in cells[1–5]. By coupling the uptake of one substrate molecule to movements of five ions—symport of three sodium ($Na^+$) ions and a proton and countertransport of one potassium ($K^+$) ion—EAATs ensure that the cytoplasmic glutamate concentration is -10 mM, which is required for neuronal functions, while the extracellular concentrations are as low as a few nM permitting rounds of synaptic signaling[6–9]. Dysfunction of EAATs leads to dysregulation of glutamate concentrations associated with central nervous system pathologies such as epilepsy, neurodegeneration, and dicarboxylic aminoaciduria[10–16].

EAATs are homotrimers composed of promoters that work independently[17–19]. TMs 1, 2, 4, and 5 of each protomer comprise the scaffold domain mediating trimerization; TMs 3, 6, 7, and 8 and helical hairpins (HPs) 1 and 2 form the substrate-binding transport domain (Supplementary Fig. 1a). EAATs transport glutamate and aspartate with similar apparent affinities[3–5,20] by an elevator mechanism, whereby the substrate-bound transport domain rotates and translocates over 15 Å across the membrane from an outward- to an inward-facing state (OFS and IFS, Supplementary Fig. 1b and Supplementary Movie 1)[21–25]. EAATs also mediate an uncoupled anion flux[4,26–29], where the conducting state is likely a transient intermediate of the elevator transition[30–33].

The EAATs' transport cycle begins with the transport domain in the OFS, where it binds sodium ions, a proton, and the transmitter. The transport domain moves into the IFS, releases the solutes, binds a potassium ion, and translocates it back into OFS for release into the extracellular space. Earlier studies suggest that cooperative binding of the transmitter, sodium ions, and protons underlies coupled transport[21,34–39]. The residues coordinating $Na^+$ and substrate and their movements upon binding are similar in OFS and IFS. Two conserved motifs, termed dancing motifs, rearrange in response to sodium binding, increasing substrate affinity (Supplementary Fig. 1c): $NMD_{366–368}$ in the unwound region of TM7 and $YE_{373–374}$ in TM7b and $DxxxDxxR_{440–447}$ in TM8 (NMD and YE/DDR motifs for short). In the absence of sodium in IFS, the motifs are in the apo configurations: the M367 sidechain points out of the substrate-binding site, and R447 salt bridges to deprotonated proton carrier residue E347 and protrudes into the site[21]. Sodium binding to the Na1 and Na3 sites induces the bound configuration: M367 swings into the binding pocket, poised to coordinate the third $Na^+$ ion in the Na2 site[40]. R447 disengages from E347 and swings out of the pocket, forming a cation-π interaction with Y373 and ready to coordinate the substrate. E347 binds a proton. The consequent binding of the transmitter and $Na^+$ in Na2, or the binding of $K^+$ ions to the apo transporter, is thought to close HP2, allowing elevator movements[21,23,34–36,38,39].

[1]Department of Physiology & Biophysics, Weill Cornell Medicine, 1300 York Ave, New York, NY 10021, USA. [2]Howard Hughes Medical Institute, Weill Cornell Medicine, 1300 York Ave, New York, NY 10021, USA. ✉e-mail: biq2001@med.cornell.edu; olb2003@med.cornell.edu

The energetic penalty for rearranging the motifs into the bound configuration underlies the cooperative binding of the transmitter and ions.

Here, we describe cryo-EM structures of human neuronal and epithelial EAAT3 (hEAAT3), which, together with our earlier structures[21], led us to overhaul the ion-coupling hypotheses of glutamate transporters. We propose that an evolutionarily conserved transient occluded intermediate outward-facing state (iOFS) ensures coupled transport by discriminating between translocation competent and incompetent transport domains. In hEAAT3, iOFS might be the only state tightly binding the transmitter and potassium. In the translocation-competent substrate/3Na$^+$/proton- and K$^+$-bound states, the zipped gating HP2 completely occludes the substrate- and ion-binding cavities. In contrast, the distorted hairpin is unzipped in the translocation-incompetent partially bound and apo states, allowing water access to the substrate- and ion-binding sites. During the return step of the transporter, the transition from iOFS to OFS is associated with the release of K$^+$, proton binding to E374, HP2 opening, and the isomerization of the dancing motifs into the bound-like configuration, priming the transporter for sodium and transmitter loading. Potassium binds to the transporter with the apo dancing motifs in place of the substrate amino group, sequestering a coordinating residue from the Na2 site. Thus, the structures explain the strictly competitive binding of the symported glutamate/3Na$^+$/H$^+$ and the counter-transported K$^+$. The structures also reveal that the evolutionary molecular innovation leading to the expansion of ion coupling in eukaryotes to include potassium ions and protons relies on glutamine to glutamate replacement in the YE/DDR motif. In archaeal transporters, glutamine stabilizes the zipped conformation of HP2, while in eukaryotes, either protonation of E374 or potassium binding is needed. Our structural analyses provide a complete description of the transport cycles of hEAAT3 and suggest energetic relationships between structural states, which couple neurotransmitter uptake to ion fluxes down their electrochemical gradients.

## Results

### Intermediate occluded states of hEAAT3 feature closed HP2 gates

Previously, we imaged hEAAT3 with mutated glycosylation sites (hEAAT3 N178T/N195T, hEAAT3g for short) in 200 mM NaCl and 1 mM aspartate. We found ~20% of protomers with bound aspartate in OFS and the remainder in IFS with bound to Na$^+$ but not aspartate[21]. These results suggested that IFS with low substrate affinity is the preferred state of the transporter and that energetically unfavorable OFS has a higher substrate affinity. We observed no glutamate (EMD: 22022) or potassium (EMD:22023) binding because of their low affinity for IFS. To facilitate imaging their complexes with the transporter, we aimed to constrain hEAAT3 in OFS by cysteine crosslinking. First, we prepared hEAAT3g with minimal cysteines to reduce spurious crosslinking. Five of six EAAT3 cysteines are not conserved; mutating them yielded MinCys EAAT3 with size exclusion chromatography (SEC) profiles similar to hEAAT3g (Supplementary Fig. 2a, b) and robust substrate uptake (Supplementary Fig. 2c, d). Even conservative mutations of the sixth highly conserved C343 in HP1 yielded inactive proteins with perturbed SEC profiles (Supplementary Fig. 2a–c). We next introduced K269C and W441C mutations in the scaffold and transport domains of MinCys EAAT3. We selected these residues because the K269C/W441C mutant is functional in cell-based assays and crosslinks in OFS[41,42]. Indeed, MinCys EAAT3 K269C/W441C shows ~20% activity compared to hEAAT3g (Supplementary Fig. 2d, e). Notably, C343 is distant and unlikely to crosslink.

After crosslinking MinCys EAAT3 K269C/W441C using Hg$^{2+}$ (EAAT3-X), we used cryo-EM to image the apo transporter in 150 mM N-methyl-D-glucamine (NMDG) chloride, sodium-bound transporter in 300 mM NaCl, neurotransmitter-bound transporter in 200 mM NaCl and 20 mM L-glutamate, and potassium-bound transporter in 300 mM KCl. Together, these structures should describe the sequential binding

of coupled solutes and visualize the potassium binding. After refining EM maps with applied C3 symmetry, we performed symmetry expansion followed by local 3D classification of the transporter protomers. In each dataset, we found protomers in OFS and a conformation resembling iOFS of archaeal homologs Glt$_{Ph}$ and Glt$_{Tk}$[38,39,43]. Thus, we obtained EM maps of the dominant and minor conformations at 2.44–2.8 Å and 2.94–3.4 Å resolution (Supplementary Tables 1–3, Supplementary Figs. 3 and 4). EAAT3-X was predominantly found in OFS in NaCl buffer and iOFS in NMDG, NaCl/glutamate, and KCl buffers (Fig. 1). Notably, we observed no K$^+$ ions bound in OFS determined in KCl buffer and assigned the state as apo (described in detail below).

To ensure symport and antiport of ions, the transport domain must undergo elevator transitions when bound to glutamate/3Na$^+$/proton or K$^+$ ions (Fig. 1a, e, f, green highlights) and not when apo or bound to sodium only (Fig. 1, pink and blue highlights). Thus, we expected translocation-incompetent species to feature the open HP2 gates seen in archaeal transporters[34,35,38,39]. Indeed, OFS structures showed wide-open hairpins (Fig. 1b–d, blue highlights), except when bound to glutamate. However, surprisingly, all iOFS structures showed closed hairpins with tips engaging the scaffold domain. The hairpins completely occluded the binding pockets in glutamate/3Na$^+$/proton- or K$^+$-bound transporters (Fig. 1e, f, green highlights) but showed solvent-accessible openings in apo and sodium-bound species (Fig. 1g, h, pink highlights).

### Glutamate mediates local and global conformational shifts

The glutamate-bound iOFS of EAAT-X is similar to the iOFS of the archaeal homologs, but the transport domain moves inward and rotates by an additional ~3 Å and 10°; we named this conformation iOFS*-Glu (Fig. 2a). The distances between the Cα atoms of C269 and C441 are 9.1 and 7.7 Å in OFS-Glu and iOFS*-Glu, respectively. Excess density, which we attribute to Hg$^{2+}$ ions, connects the sulfur atoms in both structures (Supplementary Fig. 5a, b), consistent with crosslinked cysteines. Thus, crosslinking allows transport domain movements between OFS and iOFS*. The domains show nearly identical structures (RMSD of 0.74 Å). However, the density of the loop connecting the helical arms of HP2 (HP2 tip) is weaker in OFS-Glu than in iOFS*-Glu, suggesting increased dynamics (Supplementary Fig. 6).

We found well-resolved glutamate bound to iOFS*-Glu and OFS-Glu. Highly conserved D444 and R447 in TM8 interact with the amino group and sidechain carboxylate of the transmitter, respectively, and S333 in HP1 and N451 in TM8 interact with the main-chain carboxylate (Fig. 2b). The main-chain carbonyl oxygens and amide nitrogens in HP1 and HP2 also contribute to coordination. Glutamate binds similarly to aspartate[21], except HP2b outward movement and local sidechain shifts enlarge the binding pocket slightly (Fig. 2c and Supplementary Movie 2). Interestingly, aspartate binds to hEAAT3g in an unusual rotameric state[21]. In contrast, glutamate features the same isomer as aspartate in the archaeal homologs or tsEAAT1[22,34,36]. Thus, glutamate transporters adjust the substrate-binding mode through local tuning of sidechain conformations and global HP2 movements.

The interface between the transport and scaffold domains is well-packed in OFS-Glu (Fig. 2d). In contrast, in iOFS*-Glu, a deep crevice between the domains extends from the cytoplasmic side (Fig. 2e). There are two constrictions—a tighter one near the extracellular surface and another one closer to the cytoplasm. The cytoplasmic portion of the crevice is similar to a recently proposed chloride-conducting pathway[32] but distinct from a competing model[30]. S74 in TM2, previously implicated in the selectivity of the anion channel in EAATs[44–46], contributes to the cytoplasmic constriction and hydrogen bonds to a solvent molecule in the crevice. The extracellular constriction, comprised of transport domain residues in TM7 and the HP2 tip (T364, A408, A409, and G410) and scaffold residues in TMs 2 and 4 (I67, I71, V212), has a pore radius of approximately 0.91 Å, suggesting that it is closed (Supplementary Fig. 5d, e). We found several excess density peaks in the crevice, including between the constrictions, and modeled them as water

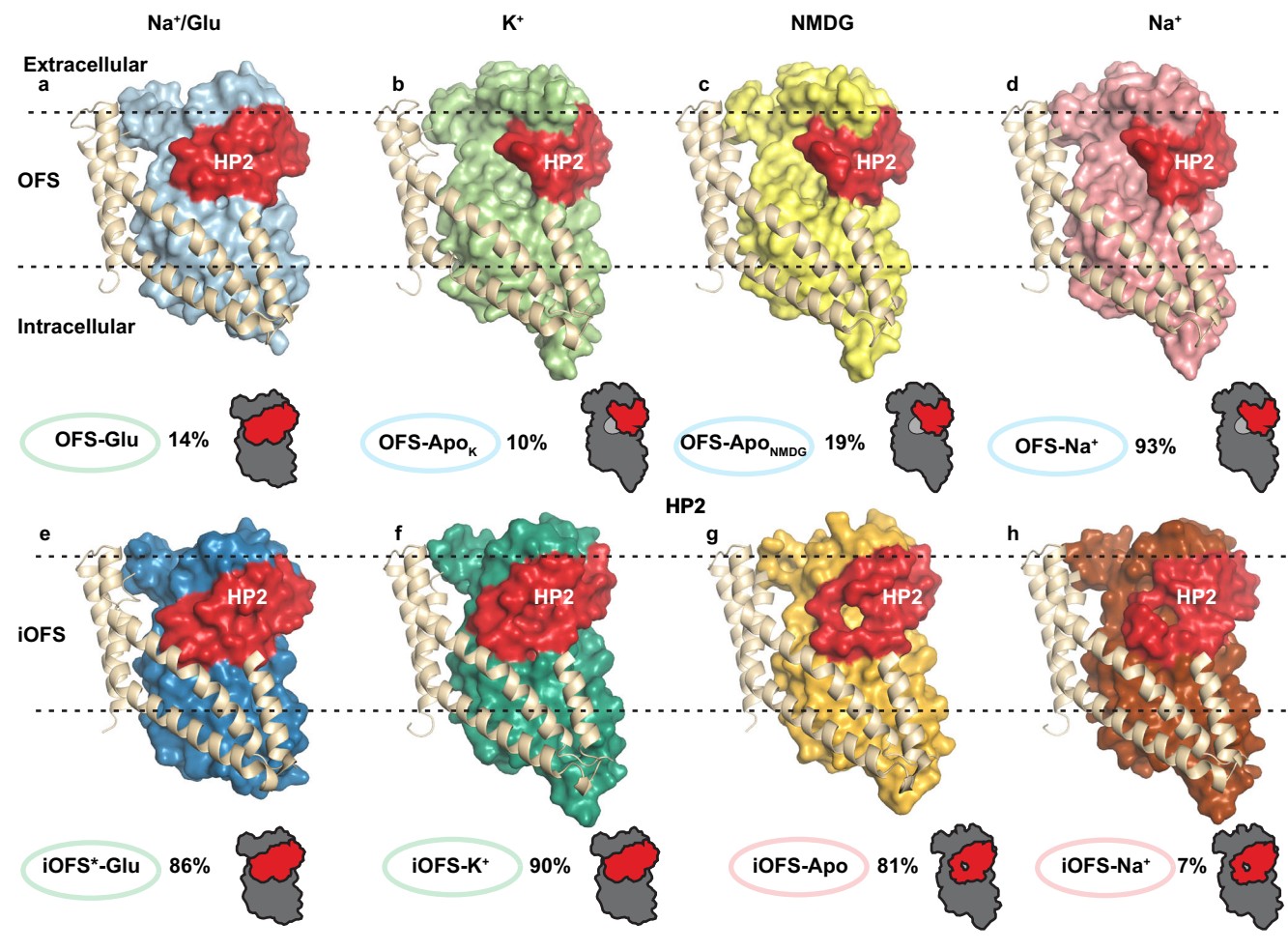

**Fig. 1 | Outward-facing and occluded intermediate states of EAAT3-X.** OFS (top) and iOFS (bottom) protomers of EAAT3-X in, from left to right, 200 mM NaCl and 20 mM L-glutamate (Na⁺/Glu), 300 mM KCl (K⁺), 150 mM NMDG chloride (NMDG), and 300 mM NaCl (Na⁺). Scaffold TMs 1, 2, and 5 and part of TM4 are shown as cartoons and colored wheat. The transport domains are shown in surface representation with HP2 colored red. The assigned conformational states and the fractions of protomers found in these states in 3D classifications are below the structures. HP2 in iOFS*-Glu, OFS-Glu, and iOFS-K⁺ fully occludes the ligand cavity (states highlighted in green, **a**, **e**, and **f**); the cavities of iOFS-Apo and iOFS-Na⁺ remain solvent-accessible (highlighted in pink, **g**, **h**); HP2 of OFS-Apo in NMDG chloride and KCl and OFS-Na⁺ are wide-open (highlighted in blue, **b–d**). The schematic diagrams under OFS-Glu, OFS-Apo_K, and iOFS-Apo represent completely occluded, wide open, and occluded with solvent-accessible openings HP2, respectively. The schematic diagrams in other figures have the same meanings.

molecules because the resolution is insufficient to distinguish water and ions (Fig. 2e). We suggest that iOFS* represents a state preceding channel opening, consistent with the more inward position of the transport domain in the proposed chloride-conducting state[32].

**Potassium binding seals the HP2 gate**

iOFS-K⁺ and iOFS-Apo imaged in the KCl and alkali-free NMDG buffers, respectively, have similar structures and feature well-resolved HP2s closed over the substrate-binding sites, although the hairpin conformations differ (Fig. 1f, g). In iOFS-K⁺, we find a strong excess density in the substrate-binding pocket, absent in iOFS-Apo, which we interpret as a bound K⁺ ion (Fig. 3a and b). Strikingly, the K⁺ ion takes the place of the substrate amino group (Fig. 3c, d). D444 coordinates both the ion and substrate, with T448 and the carbonyl oxygen atoms of residues in HP2 (408, 409, and 411) and HP1 (331) contributing to ion coordination (Fig. 3a). Consistently, the D444S/T448A EAAT3 mutant shows impaired potassium binding and transports dicarboxylates instead of glutamate[47]. Mutations of D444 to any amino acid, including glutamate, abolish transport currents, and the T448A mutation alone decreases uptake by 80%[48].

Potassium binding at this site is consistent with earlier proposals based on modeling[49] and the crystallographic studies of thallium binding to archaeal Glt_Ph[35] and rubidium binding to tsEAAT1[23]. In rubidium-bound tsEAAT1 in OFS, HP2 remained open, and ion coordination was incomplete. Here, we observe complete octahedral ion coordination with distances between the ion and the coordinating oxygens between 2.5 and 3.6 Å, within the expected range[50]. Furthermore, the efficient occlusion of the ion by compact closed HP2 strongly supports the site as the bona fide potassium binding site of EAATs. We see no evidence of K⁺ ions at other proposed binding sites, including Na1 and sites between Na1 and Na3[51,52].

Strikingly, K⁺ and Na2 sites are mutually exclusive. In iOFS*-Glu, carbonyl oxygens of residues in the last turn of the HP2a helix, 405, 406, and 408, coordinate Na2. In iOFS-K⁺, these residues unwind, while residues 408, 409, and 411 in the tip form a turn-of-a-helix-like structure, coordinating the K⁺ ion (Fig. 3c, d). The overlapping binding sites for glutamate/Na2 and potassium suggest that the solutes compete for binding, consistent with the antiport mechanism.

Notably, archaeal transporters do not couple to potassium countertransport or bind potassium tightly[51]. This is surprising because

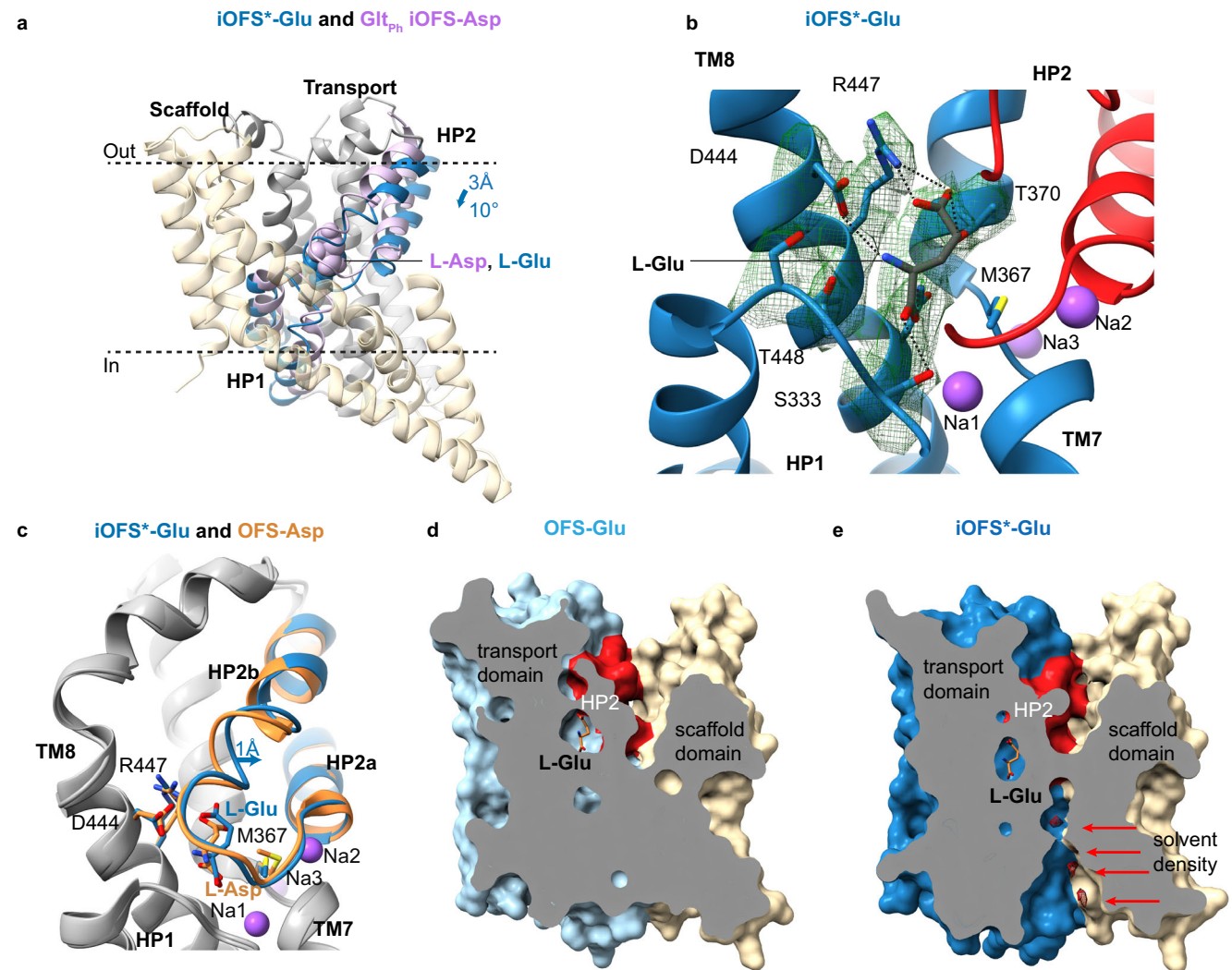

**Fig. 2 | Glutamate binding to EAAT3. a** Superposition of EAAT3-X iOFS*-Glu and aspartate-bound GltPh iOFS-Asp (PDB ID: 6UWL) aligned on the oligomerization regions of the scaffold domains (wheat). The transport domains are colored gray, and HP1 and HP2 are blue in EAAT3-X and light violet in Glt_Ph. Bound L-Asp and L-Glu are shown as light violet and blue spheres. **b** Glutamate coordination by EAAT3-X. Bound substrate and coordinating residues are shown as sticks and colored by atom type. The green mesh is the EM density contoured at 5σ. Dashed lines represent hydrogen bonds between L-Glu and the coordinating residues.

**c** Transport domains of aspartate-bound hEAAT3g OFS (OFS-Asp, PDB ID: 6X2Z) and iOFS*-Glu superimposed on their cytoplasmic halves (residues 314−372 and 442−465). Transport domains are colored gray with HP2, substrate, and coordinating residues colored orange and blue, respectively. Surface representation of OFS-Glu (**d**) and iOFS*-Glu (**e**), sliced through the substrate-binding site and the domain interface. Arrows emphasize the observed solvent EM densities, shown as red mesh and contoured at 5σ. OFS-Glu and iOFS-Glu* were aligned on the scaffold domain residues 17−85 and 132−272.

thallium bound to Glt_Ph and potassium bound to EAAT3 share coordination by D444 and T448. However, in archaeal transporters, glutamine replaces E374 (EAAT3 numbering), and R447 in Apo Glt_Ph, lacking its interaction partner, extends further into the substrate-binding site where it might electrostatically interfere with potassium binding. Furthermore, HP2 does not form a turn-of-a-helix structure in the archaeal transporter, and residues 409, 411, and 412 coordinate the ion in an extended loop configuration. The ability to form a helical turn in HP2 seems important for K⁺ binding because EAATs share a conserved coordinating double alanine motif $AA_{408-409}$ with a high helical propensity. Prokaryotic homologs do not show such conservation.

**Mechanism of sodium, proton, and glutamate symport and potassium antiport**

To ensure coupled transport and prevent ion gradient dissipation, iOFS-K⁺, and iOFS*-Glu should be able to translocate into the IFS, but iOFS-Apo and iOFS-Na⁺ should not. What are the structural correlates distinguishing translocation-competent and translocation-

incompetent states? iOFS-Apo is structurally similar to iOFS-K⁺ (RMSD of 0.52 Å for the transport domains; Fig. 4a); both feature apo configurations of the NMD and YE/DDR motifs (Fig. 4b). Analogously, iOFS-Na⁺ is similar to iOFS*-Glu (RMSD of 0.72 Å, Fig. 4c). The NMD motif of iOFS-Na⁺ is in the bound configuration with the Na1 and Na3 sites occupied (Fig. 4d). All structures have closed HP2s (Fig. 1e-h). However, in translocation-incompetent iOFS-Apo and iOFS-Na⁺, HP2 residues coordinating K⁺ and Na2 and residues 414−417 at the start of the HP2b helix unwind and form an enlarged loop with an opening in the middle (Fig. 1g, h, states highlighted in pink), resembling an unzipped purse. The potassium binding to iOFS-Apo and glutamate to iOFS-Na⁺ zip HP2s, increasing their helicity and decreasing dynamics, as evidenced by the better-defined HP2 EM density of iOFS-K⁺ compared to iOFS-Apo (Supplementary Fig. 6).

A comparison of EAATs to archaeal homologs, which do not couple to potassium countertransport and can undergo elevator transition in Apo states, suggests that zipping HP2 is necessary for transport domain translocation. In archaeal transporters, HP2b

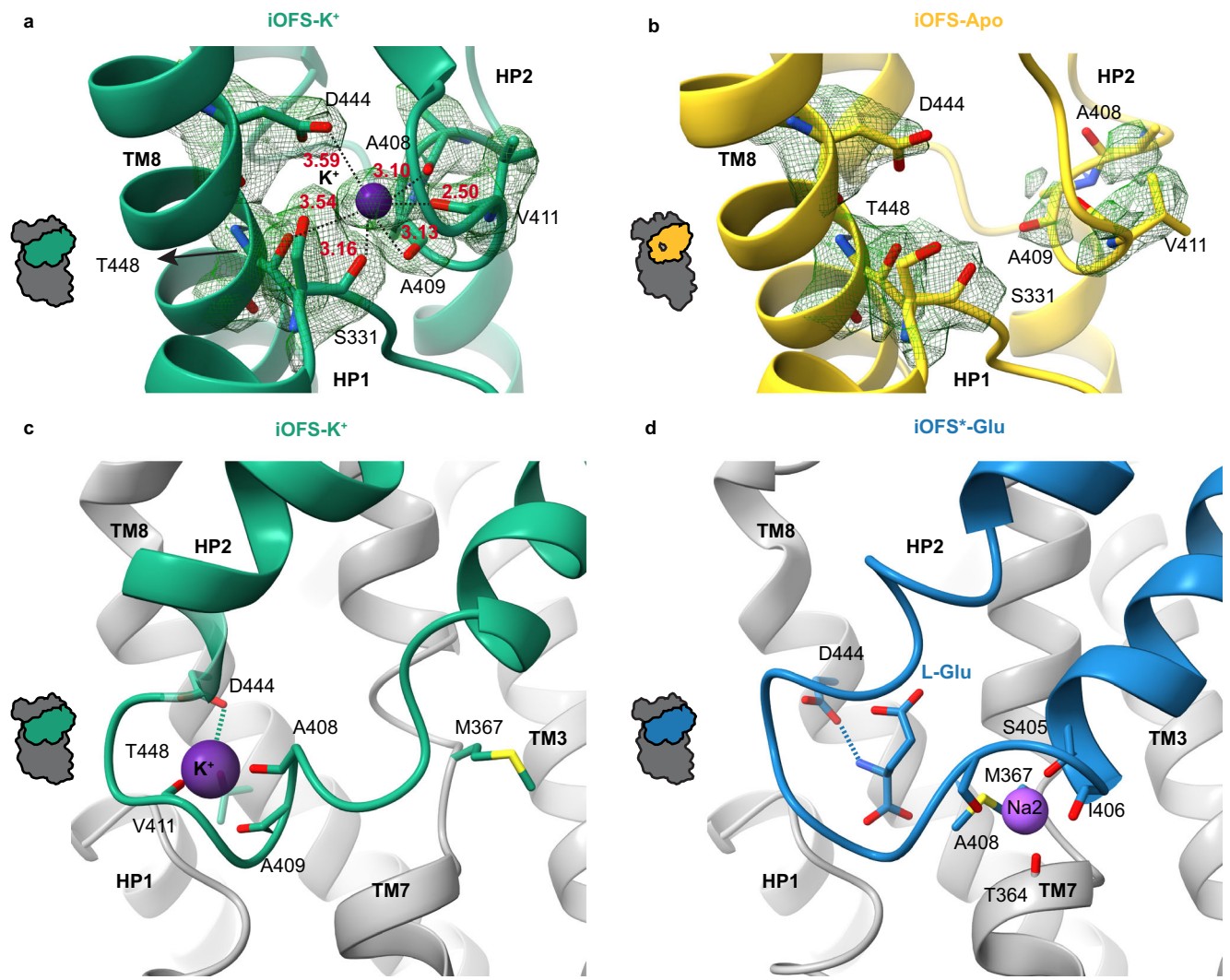

**Fig. 3 | Potassium binding. a, b** The substrate-binding site in iOFS-K⁺ (**a**) and iOFS-Apo (**b**). EM density, contoured at 5σ, is shown as a green mesh. Dotted lines emphasize the interactions between potassium and the coordination oxygen atoms with distances listed in Å. **c, d** Potassium (**c**) and Na2 (**d**) binding sites with coordinating moieties emphasized as sticks. Dashed lines emphasize the shared coordination by D444 of K⁺ and ʟ-Glu.

remains helical even without K⁺ ions. Their YE/DDR motifs feature E374Q replacement and glutamine hydrogen bonds to the carbonyl oxygen of G415 in HP2b, stabilizing the helix. In contrast, deprotonated E374 of apo EAAT3 is engaged with R447 and cannot stabilize the helix, necessitating K⁺ binding to zip the hairpin (Supplementary Fig. 7a–c).

Sodium binding to Na1 and Na3 sites restructures the NMD motif, leading to changes in the tilt and rotation of TM7b and disengaging E374 from R447 in iOFS-Na⁺ (Fig. 4d). As a consequence, K⁺ binding to iOFS-Apo and glutamate binding to iOFS-Na⁺ stabilize zipped HP2s in different positions, with the hairpin shifting by 2 Å toward the cytoplasm and rotating by 5° in iOFS*-Glu compared to iOFS-K⁺ (Fig. 4e). In iOFS*-Glu, the E374 sidechain carboxyl oxygen positions were within 3 Å of the carbonyl oxygen of G415 in HP2b (Fig. 4f), suggesting that the two share a proton, as previously proposed[23]. The formed hydrogen bond likely stabilizes the HP2b helix, mimicking interactions in the archaeal transporters (Supplementary Fig. 7c, d). Thus, E374 protonation is required for glutamate-mediated HP2 zipping. The observations that the E374Q mutation and mutations of Y373 and R447 abolished potassium coupling and concentrative glutamate transport in EAATs[53–56] support the crucial importance of the YE/DDR motif in coupling.

Comparing iOFS-Na⁺ to iOFS-Apo structures, we observed that sodium binding led to M367 in the NMD motif shifting into the binding pocket and R447 in the YE/DDR motif shifting out (Fig. 4b, d). R447 repositioning is necessary to coordinate glutamate[21,35,36,38,39], while M367 movement allows the extension of the HP2a helix necessary to coordinate Na2 (Supplementary Fig. 7e, f). Thus, in contrast to iOFS-Na⁺, iOFS-Apo cannot bind and translocate glutamate, preventing the dissipation of the glutamate gradient. On the other hand, simple modeling shows that M367 must vacate the binding pocket, as it does in iOFS-Apo, to make space for the turn-of-the-helix in the HP2 tip coordinating the K⁺ ion. Thus, iOFS-Na⁺ cannot bind potassium, preventing aberrant sodium/potassium symport.

**Protons promote OFS, gating, and sodium binding**

In contrast to iOFS-Na⁺, OFS-Na⁺ shows wide-open HP2 and a fully solvent-exposed substrate-binding site (Fig. 1d and Fig. 5a). R447 interacts with Y373, already in place for glutamate coordination (Fig. 5b). Surprisingly, when we examined the OFS conformations imaged in the NMDG chloride and KCl buffers, we found that they closely resembled OFS-Na⁺ with wide-open HP2 gates (Fig. 1b, c and Fig. 5a). We observed no density in the K⁺-binding site, suggesting that potassium did not bind to OFS even though it was present at 300 mM. We term these nearly identical structures (RMSD 0.57 Å) OFS-Apo_NMDG and OFS-Apo_K and refer to them collectively as OFS-Apo for short. Strikingly, the YE/DDR motifs of OFS-Apo are in the

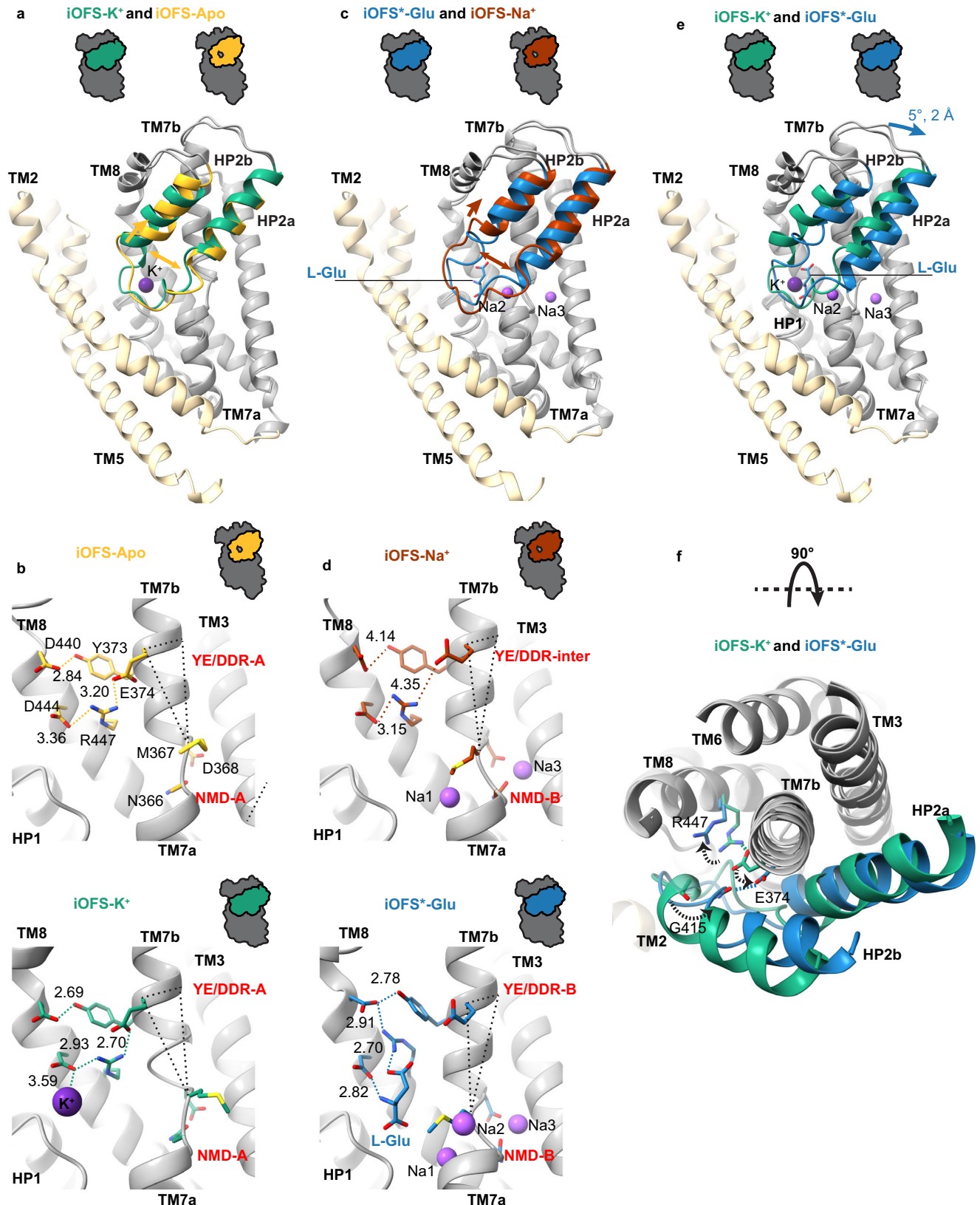

bound configuration, while their NMD motifs are in a configuration intermediate between apo and bound conformations (Fig. 5c). Inspection of the Na3 sites in OFS-Apo$_{NMDG}$ and OFS-Apo$_K$ revealed distorted geometries of the ion-coordinating residues, including shifted N366 sidechains (Fig. 5c). We found excess nonprotein densities at the sites, likely due to water molecules (Supplementary

Fig. 8a). Similarly, slight shifts of the coordinating N366 and N451 carbonyl oxygens distort the Na1 sites, showing no significant excess density peaks (Fig. 5c and Supplementary Fig. 8b).

We then examined the PROPKA-predicted pKa values and solvent accessibilities of the proton carrier residue E374 in all our structures (Supplementary Table 4). In iOFS*-Glu and OFS-Glu, E374 was

**Fig. 4 | K+- and glutamate-mediated zipping of HP2. a, c, e** Superimposed structures of iOFS-K+ and iOFS-Apo (**a**), iOFS\*-Glu and iOFS-Na+ (**c**), and iOFS-K+ and iOFS\*-Glu (**e**). HP2 is colored green in iOFS-K+, yellow-orange in iOFS-Apo, blue in iOFS\*-Glu, and brown in iOFS-Na+. The protomers are superimposed on the cyto-plasmic halves of the transport domain (residues 314–372 and 442–465). The scaffold domains are shown for one protomer for clarity and colored wheat. Yellow-orange, brown, and blue arrows indicate HP2 movements from iOFS-K+ to iOFS-Apo, iOFS\*-Glu to iOFS-Na+, and iOFS-K+ to iOFS\*-Glu. The K+ and Na+ ions are shown as dark purple and purple spheres. **b, d** The NMD and YE/DDR motifs in iOFS-Apo (**b**, top), iOFS-K+ (**b**, bottom), iOFS-Na+ (**d**, top), and iOFS\*-Glu (**d**, bottom). The motifs are labeled in red and are in apo (-A), bound (-B), or intermediate (-inter) conformations. Dotted triangles connect M367, E374, and the edge of the TM7b helix. Changes in the triangle reflect rotation and tilt changes of TM7b observed upon the transition from the apo to the bound conformation of the NMD motif. Colored dashed lines mark key interactions with distances shown in Å. **f** Extracellular view of iOFS-K+ and iOFS\*-Glu superposition. Movements of E374, G415, and R447 are shown as dotted arrows.

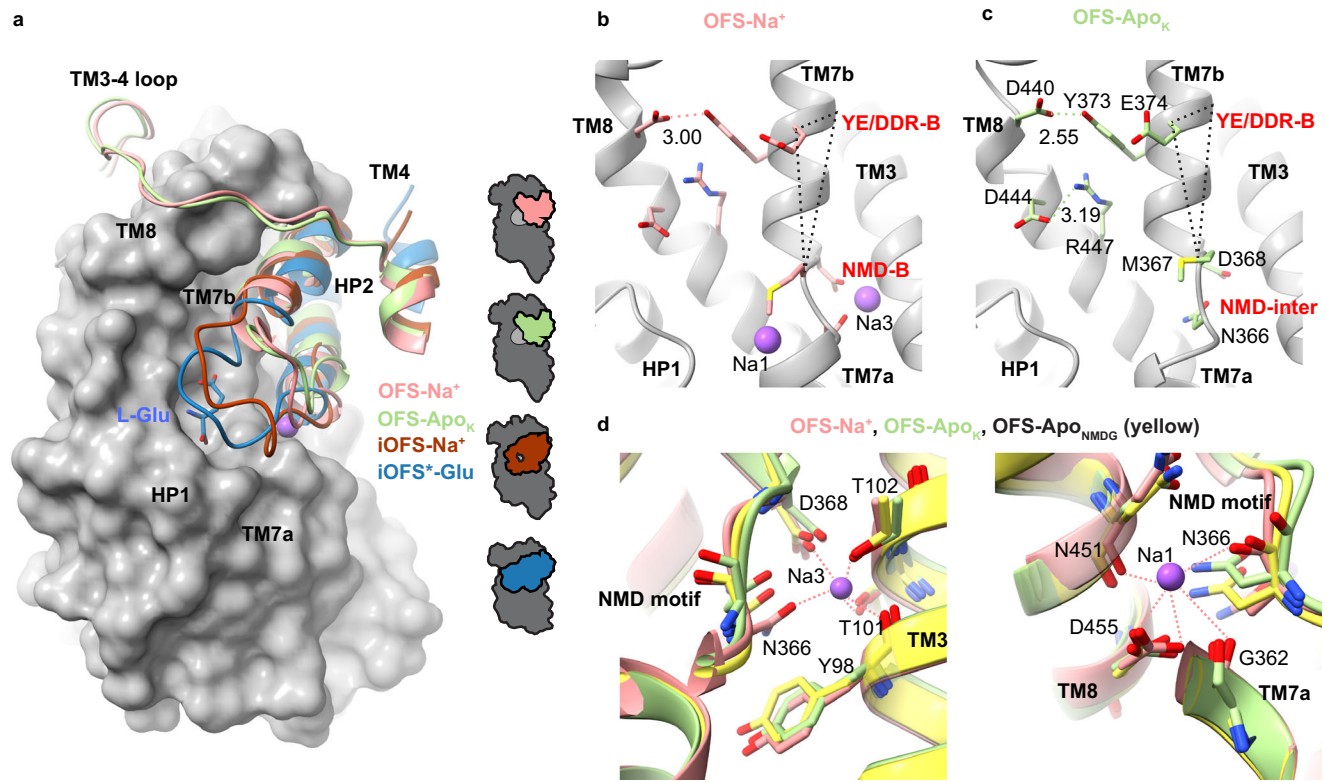

**Fig. 5 | HP2 opening and the NMD and YE/DDR motif structures in OFS.**
**a** Superposition of the transport domains of iOFS\*-Glu (blue), iOFS-Na+ (brown), OFS-Na+ (pink), OFS-Apo_K (light green), aligned on the cytoplasmic half of the transport domain (residues 314–372 and 442–465). The transport domain of iOFS\*-Glu, except HP2, is shown as a gray surface. HP2 is shown in cartoon representation. The TM3–4 loop and a section of TM4 are also shown for the OFS structures. **b, c** The NMD and YE/DDR motifs in OFS-Na+ and OFS-Apo_K. The labeling scheme is the same as in Fig. 4. **d** Comparison of the Na3 (left) and Na1 sites (right) in OFS-Na+ (pink), OFS-Apo_K (light green), and OFS-Apo_NMDG (yellow). Pink dashed lines represent the interactions between Na+ ions and the coordinating atoms.

completely occluded from the solution. The predicted pKa-s of 8.4 and 8.9, respectively, suggest that the residue paired with G415 is proto-nated, as expected for a fully loaded transporter. In iOFS-K+, E347 is also buried and paired with R447. Its pKa of 5.5 suggests that E347 is deprotonated, as expected for potassium countertransport. In iOFS-Apo and iOFS-Na+, E374 is solvent-accessible through the unzipped HP2 and likely deprotonated with pKa-s of 6.5 and 7.1, respectively. In contrast, in OFS-Apo and OFS-Na+, E374 is buried and proximal to G415. It is likely protonated with pKa-s of 9.2, 10.0, and 7.6 for OFS-Apo_K, OFS-Apo_NMDG, and OFS-Na+, respectively.

Collectively, E374 is deprotonated in all iOFS structures, except iOFS\*-Glu, and protonated in all OFS structures. Despite inherent uncertainties of the predicted pKa values, we suggest that proton binding to E374 favors OFS with bound-like conformations of the dancing motifs and wide-open HP2, ready to bind sodium ions and substrate.

## Discussion

EAAT3 is a neuronal and epithelial subtype of glutamate transporters, the best functionally characterized of all EAATs[1,6,48,54,56–64]. Our eight hEAAT3 structures provide the basis for substrate recognition and selectivity and the thermodynamic coupling of substrate uptake to transmembrane movements of three sodium ions, a proton, and a potassium ion.

EAATs transport glutamate and aspartate with similar micromolar affinities. In contrast, extensively studied archaeal transporters bind aspartate with nanomolar affinity, ten thousand times tighter than glutamate. This difference in substrate selec-tivity is remarkable because the substrate-binding sites are nearly identical in all structures. Moreover, hEAAT3 accommodates glutamate or aspartate upon sub-Angstrom shifts of the coordi-nating residues and a subtle movement of the HP2b arm, reg-ulating the pocket volume (Supplementary Movie 2). Recent studies on Glt_Ph suggest that disrupting HP2 packing against the transport domain can reduce substrate affinity[65,66]. Therefore, we speculate that the tight protein packing in hyperthermophilic Glt_Ph accounts for its high affinity for aspartate and explains the large energetic penalty to enlarge the binding pocket for glutamate that requires movements of HP2 and altered interac-tions with the transport domain. In contrast, looser packing in

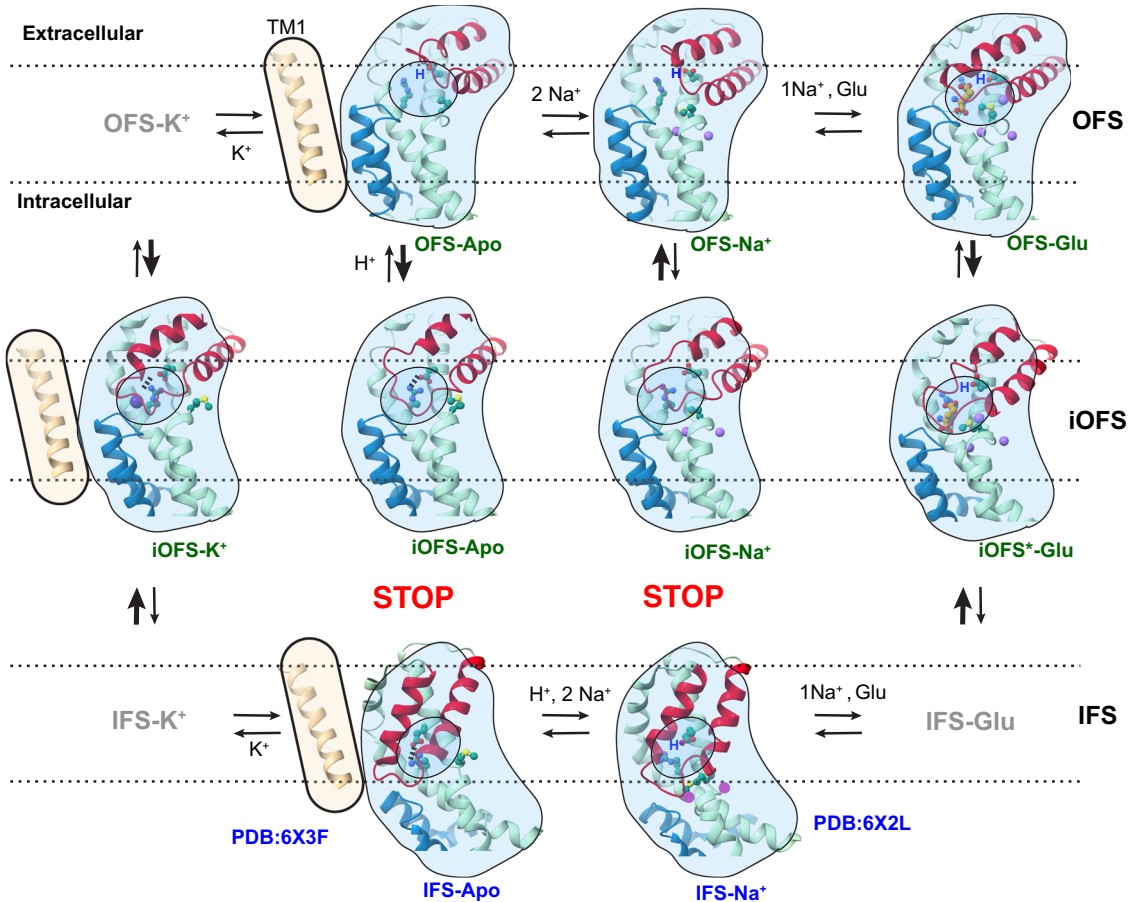

**Fig. 6 | Molecular mechanism of EAAT3.** Only TM1 (wheat), HP1 (blue), HP2 (red), and TMs 7 and 8 (light green) of a single EAAT3 protomer are shown for clarity. The substrate glutamate (orange), M367 in the NMD motif, and E374 and R447 in the YE/DDR motif (dark green) are shown as ball-and-stick models. The K[+] and Na[+] ions are shown as dark purple and purple spheres, respectively. Blue "H" marks the proton bound to E374. The dashed lines indicate the approximate position of the lipid bilayer. The high-energy unresolved states are marked in gray. The previously resolved IFS states are labeled in blue with PDB codes; the newly captured states are labeled in green. The conformational changes that are unlikely to happen are marked with red STOP signs.

mammalian EAATs allows them to adapt their binding pockets to aspartate and glutamate with small energetic penalties and transport both with similar moderate affinities.

Crosslinked EAAT3-X populates OFS and occluded iOFS or iOFS* (Figs. 1 and 6). Remarkably, the state is conserved from archaea to humans[38,43], suggesting its functional significance. In archaeal Glt$_{Ph}$ reconstituted into nanodiscs, ~10% of the protomers spontaneously assume iOFS. In EAAT3-X, it is possible that crosslinking stabilizes iOFS or iOFS* because we observed only OFS in uncrosslinked aspartate-bound hEAAT3g[21]. Thus, it might be a transient state, lowly populated in the wild-type transporter. HP2s is closed in all EAAT3-X iOFS conformations. In contrast, OFS-Na[+] and OFS-Apo feature fully open HP2-s, and only transmitter binding can close the gate. hEAAT3g in the IFS, bound to Na[+] ions or apo, also pictures an open HP2[21]. In this regard, EAAT3 differs from Glt$_{Ph}$, where the hairpins close in the apo IFS and OFS. This difference might arise from stronger interactions between open EAAT3 HP2 and scaffold TMs 2 and 5 in IFS or the TM3-4 loop in OFS, favoring open conformations. In iOFS, the hairpins disengage from the TM3-4 loop, and their tips interact with TMs 2 and 5 instead as the transport domain moves inward. We suggest that the discrimination between the translocation-competent and incompetent states, which is at the heart of ion-coupling mechanisms, takes place at this point. The zipped glutamate/3Na[+]/H[+]- or K[+]-bound transport domains (Fig. 1, green highlights) can move beyond iOFS to IFS, while the unzipped domains of apo and sodium-only bound transporter (pink highlights) cannot. The conclusion that translocation-

incompetent transport domains feature mostly closed HP2s contrasts sharply with earlier hypotheses that their HP2s would be wide-open as in OFS (blue highlights)[35,38,39,67–69].

Our cryo-EM image analysis approach has previously produced state populations in good agreement with solution measurements[70]. Therefore, we interpret the populations observed in 3D classifications with some confidence. We observed that iOFS and OFS populations shifted from ~7 and 93%, respectively, when the protein was bound to sodium to 86 and 14%, respectively, when it bound glutamate. Therefore, the equilibrium constant between OFS and iOFS increases from $K_{eq,Na}$ of ~0.076 to $K_{eq,Glu}$ of ~6.1 when glutamate binds. The ratio of the binding constants in iOFS and OFS ($K_{B,iOFS}$ and $K_{B,OFS}$) equals the ratio of the equilibrium constants: $\frac{K_{B,iOFS}}{K_{B,OFS}} = \frac{K_{eq,Glu}}{K_{eq,Na}}$. Thus, we estimate that iOFS has ~80 times higher affinity for glutamate than OFS. Notably, we observed little substrate binding to IFS in 200 mM Na[+] and 20 mM aspartate by cryo-EM (EMD:22024), suggesting that IFS binds substrate very weakly. Therefore, iOFS* has the highest affinity for the transmitter among all states. Furthermore, iOFS is the only state that binds K[+] ions; neither IFS nor OFS bind even in 250–300 mM KCl (EMD: 22023, and Fig. 5). We suggest that iOFS accounts for the measured ~10 μM[1,61] and 15 mM[6,71] $K_m$ of glutamate and potassium transport, respectively. The propensity of HP2 to close in iOFS might explain why the state binds these solutes tightly.

In contrast, Na[+] ions bind ~50-fold weaker to iOFS than OFS, based on the equilibrium constant between iOFS and OFS of ~4.2 for the apo transporter. Considering the measured sodium Km of 20 mM[23], we

estimate iOFS sodium affinity at ~1 M. Protons also bind approximately a thousand times tighter to OFS, based on the predicted E374 pKa values of iOFS-Apo and OFS-Apo.

Overall, we propose the following description of the transport cycle of EAAT3. OFS-Apo features an open HP2, protonated E374, and bound-like NMD and YE/DDR motifs (Fig. 6). Na⁺ ions bind with high affinity to OFS-Apo, and OFS-Na⁺ might be the resting state in cells under physiologic conditions. OFS-Apo and OFS-Na⁺ can visit iOFS when transiently deprotonated but cannot progress to IFS because of unzipped HP2s. The neurotransmitter and Na2 bind to OFS-Na⁺, promoting HP2 closure and transition into iOFS*-Glu and IFS-Glu. Glutamate rapidly dissociates from the low-affinity IFS, followed by Na⁺ ions and the proton, yielding IFS-Na⁺ and IFS-Apo, which are open and unable to translocate into iOFS and OFS. K⁺ ion replaces the substrate leading to HP2 closure and a rapid transition into iOFS-K⁺. The formation of the transient K⁺-bound IFS state might be the rate-limiting step of the transport cycle, consistent with the functional data[61]. E374 protonation triggers the transition into OFS and K⁺ release, completing the cycle.

In conclusion, we present a complete structural description of the hEAAT3 transport cycle, highlighting the critical role of the occluded intermediate state. We also provide insights into the coupling mechanisms, energetics, substrate selectivity, and transport kinetics. The same structural states and coupling mechanisms likely exist in other glutamate transporter subtypes, although the relative state energies, populations, and kinetics may vary.

## Methods

### Protein expression and purification

Codon-optimized full-length human EAAT3 with N178T and N195T mutations (hEAAT3g)[21] was cloned into a modified pCDNA 3.1 (Invitrogen) plasmid with an N-terminal Strep-II tag followed by GFP and the PreScission protease site. Cysteine residues 9, 100, 158, and 219 were mutated to alanines and C256 to tryptophan found in other EAAT subtypes, yielding the MiniCys EAAT3 construct. C343 in MinCys EAAT3 was mutated to alanine, glycine, serine, threonine, and valine. The double cysteine mutation K269C/W441C was introduced in Mini-Cys EAAT3 for crosslinking. All mutations were prepared by site-directed mutagenesis (Agilent). The sequences of primers were provided in the source data file. One liter human embryonic kidney (HEK) 293 F cells cultured in Freestyle 293 medium (Gibico) to a density of approximately 2.5 million cells/ml were transfected using 3 mg plasmid using poly-ethylenimine (PEI, Polysciences) with a 1:3 plasmid to PEI weight ratio. Transfected cells were diluted with a liter of fresh Freestyle 293 medium 6 h after transfection. Twelve hours later, 2.2 mM valproic acid sodium (Sigma-Aldrich) was added to the cells to boost protein expression. Approximately 48 h after transfection, the cells were collected by centrifugation at 4000g for 10 min at 4 °C. The cells were resuspended in lysis buffer containing 50 mM Tris-Cl pH 8.0, 1 mM L-aspartate, 1 mM EDTA, 1 mM tris(2-carboxyethyl) phosphine (TCEP), 1 mM phenylmethylsulfonyl fluoride (PMSF), and a 1:200 dilution of protease inhibitor cocktail (P8340, Sigma-Aldrich). Resuspended cells were disrupted using an EmulsiFlex-C3 cell homogenizer (Avestin). The cell debris was removed by centrifugation at 10,000g for 15 min at 4 °C, and membrane pellets were collected by ultracentrifugation at 186,000g for 1 h at 4 °C. The membrane pellets were homogenized in lysis buffer supplemented with 200 mM NaCl and 10% (v/v) glycerol (solubilization buffer) and incubated with 1% dodecyl-β-D-maltopyranoside (DDM; Anatrace) and 0.2% cholesteryl hemisuccinate (CHS; Sigma-Aldrich) at 4 °C overnight. Insoluble material was removed by ultracentrifugation at 186,000g for 1 h at 4 °C, and the supernatant was incubated with Strep–Tactin Sepharose resin (GE Healthcare) for 1 h at 4 °C. The resin was washed with eight column volumes of wash buffer containing 50 mM Tris-Cl pH 8.0, 200 mM NaCl, 0.06% glyco-diosgenin (GDN; Anatrace), 1 mM TCEP, 5% glycerol,

and 1 mM L-aspartate. Protein was eluted with four column volumes of the wash buffer supplemented with 2.5 mM D-desthiobiotin (elution buffer). The N-terminal Strep II and eGFP tags were cleaved by incubating the protein with homemade PreScission protease at a 40:1 protein-to-protease ratio overnight at 4 °C. Eluted protein was further purified by size exclusion chromatography (SEC) using a Superose 6 Increase 10/300 column (GE Healthcare) preequilibrated with the desired buffer.

To prepare crosslinked EAAT3-X with 20 mM L-glutamate, the protein was first purified by SEC in a buffer containing 20 mM HEPES-Tris pH 7.4, 200 mM NaCl, 0.01% GDN, and 1 mM L-aspartate. Peak SEC fractions were pooled, concentrated to ~0.5 mg/ml, and incubated with HgCl₂ at a 1:20 protein to Hg²⁺ molar ratio for 15 min at room temperature. L-aspartate was used during purification and crosslinking because it binds tighter to the transporter than L-glutamate, providing better protein stabilization. Crosslinked EAAT3-X was further exchanged into a buffer containing no sodium or aspartate by SEC in 20 mM HEPES-Tris pH 7.4, 100 mM choline chloride, and 0.01% GDN. The peak fractions were supplemented with L-glutamate by diluting ~1,000-fold into a buffer containing 20 mM HEPES-Tris pH 7.4, 200 mM NaCl, 20 mM L-glutamate, and 0.01% GDN and concentrated using 100 kD MWCO concentrators (Amicon). EAAT3-X protein samples in 300 mM KCl and 150 mM N-methyl-D-glucamine (NMDG) chloride were prepared by SEC in buffers containing 20 mM HEPES-Tris pH 7.4, 0.01% GDN, and 300 mM KCl or 150 mM NMDG chloride. To prepare EAAT3-X in 300 mM NaCl, protein in 150 mM NMDG buffer was diluted ~1,000-fold into a buffer containing 20 mM HEPES-Tris pH 7.4, 300 mM NaCl and 0.01% GDN. Notably, we prepared NMDG and Na⁺ samples using the same purified crosslinked protein.

### Fluorescence size exclusion chromatography

For fluorescence-detection size exclusion chromatography (FSEC), 10 mL of cells were transfected with the desired constructs, as described above. After 48 h, 5 mL of cells were centrifuged at 4,000 g for 10 min at 4 °C. The cell pellets were resuspended in 400 μl solubilization buffer and incubated with 2% DDM and 0.4% CHS at 4 °C for 1 h. The lysates were centrifuged at 15,000g for 30 min at 4 °C. The supernatants were filtered using a centrifuge tube filter (Corning Costar Spin-X) and diluted 2-fold with FSEC buffer containing 50 mM Tris-Cl pH 8.0, 200 mM NaCl, 0.03% DDM, 0.006% CHS, 1 mM TCEP, and 1 mM L-aspartate. Fifty microlitre samples were injected into a Superose 6 Increase 10/300 column preequilibrated with FSEC buffer. GFP fluorescence was monitored by a fluorescence detector (Shimadzu RF-20A) in-line with AKTA pure 25 M (GE Healthcare).

### Radiolabeled aspartate uptake into cells

For uptake, 40 mL of cells were transfected with the desired constructs, as described above. After 48 h, the cells were imaged using a fluorescence microscope (BZ-X series, Keyence) according to the user manual. The cells were then pelleted at 200g for 10 min at room temperature. The cell pellets were washed with cellular uptake resting buffer containing 11 mM HEPES/Tris, pH 7.4, 140 mM choline chloride, 4.7 mM KCl, 2.5 mM CaCl₂, 1.2 mM MgCl₂, and 10 mM D-glucose. The cells were pelleted again and resuspended in a resting buffer at 50 million cells/ml density. The cells were diluted 10-fold into uptake buffer containing 11 mM HEPES/Tris, pH 7.4, 140 mM NaCl, 4.7 mM KCl, 2.5 mM CaCl₂, 1.2 mM MgCl₂, and 10 mM D-glucose or the resting buffer as a control and incubated for 2 min at 37 °C. Uptake was initiated by adding 1 μM [3H] L-aspartate and 19 μM L-aspartate. Two-hundred microlitre samples were transferred at selected time points into 2 mL of cold resting buffer to stop the reaction. The samples were filtered using 0.8 μm filters (MF-Millipore) and washed with 6 ml of cold resting buffer. The filters were transferred into scintillation liquid, and the retained radioactivity was quantified using a scintillation counter (Beckman Coulter LS6500). The cellular uptake assays were performed

3 times for each construct using independently prepared samples, each with a technical duplicate.

## Cry-EM data acquisition

Purified protein samples (3.5 µl) at 4–6 mg/ml were applied to glow-discharged QUANTIFOIL R1.2/1.3 holey carbon-coated 300 mesh gold grids (Quantifoil, Großlöbichau, Germany). Grids were blotted for 3 s at 4 °C and 100% humidity and plunge-frozen into liquid ethane using FEI Mark IV Vitrobot (FEI, part of Thermo Fisher Scientific, Hillsboro, OR). The dataset for EAAT3-X with 20 mM L-glutamate was collected using Leginon[72] at the New York Structural Biology Center (NYSBC). The datasets for crosslinked EAAT3-X with 300 mM KCl, 150 mM NMDG, and 300 mM NaCl were collected using Leginon[72] at NYU Langone Health's Cryo-Electron Microscopy Laboratory. The detailed parameters of cryo-EM data collection are provided in Supplementary Table 1.

## Image processing

All the datasets were processed using a combination of Relion 3.1.0[73] and cryoSPARC v3.2.0[74] software packages. Movie drift correction was performed using Motioncor2[75], and the CTF parameters of the micrographs were estimated using CTFFIND[76] in Relion 3.1.0.

For the EAAT3-X in 20 mM L-Glu dataset, 8,511,485 particles were auto-picked from 6361 micrographs by Laplacian-of-Gaussian (LoG). The particles were extracted using a box size of 128 pixels with 2× binning and imported into cryoSPARC for 2D classification. A total of 1,087,381 particles showing secondary structure features were selected and subjected to 3 consecutive rounds of ab initio reconstructions. A total of 88,094 particles producing a volume with the most features were refined to 6.2 Å by nonuniform refinement[77] (hereafter, NUR), applying C1 symmetry. Separately, the same 8,511,485 particles were subjected to 2D cleaning, during which nonprotein junk particles were removed, yielding 6,867,502 particles. These were subjected to a round of heterogeneous refinement using one good volume from NUR and eight decoy noise volumes. For this and other datasets, the noise volumes were generated by running 1–10 iterations of ab initio reconstructions using particles after 2D cleaning. This procedure, termed heterogeneous refinement for cleaning (HRC), sorts informative particles from noise. The resulting 1,726,968 particles were refined to 6.3 Å by NUR in C1. The particles were reimported into Relion using PyEM[78] and, after unbinning, re-extracted using a 256-pixel box size. These particles were reimported into cryoSPARC and subjected to a round of HRC. The resulting 682,971 particles were refined to 3.35 Å by NUR with C3 symmetry. The refined particles underwent two rounds of polishing in Relion, HRC, and NUR. The final map was refined to 2.8 Å with C3 symmetry using 496,972 particles. Symmetry expansion and local classification without alignment with a mask applied over the transport domain yielded 14% protomers in a different conformation. These particles produced a 3.42 Å resolution map after local refinement with a mask applied over the protomer and postprocessing in Relion. Because protomers in the trimer work independently, the minor conformations come from trimers, with the other protomers in the major conformations. To obtain the highest-resolution maps of the major conformations, we used all trimers, including those that contain protomers in the minor conformation, and aligned them with the imposed C3 symmetry. The contributions from the minor conformations are averaged out in the final maps. This approach produced higher-resolution maps than refining the major conformation protomers separately.

For the EAAT3-X in 300 mM KCl dataset, 3,502,663 particles were auto-picked from 5116 micrographs by LoG. Unbinned particles extracted with a 300-pixel box size were imported into cryoSPARC for 2D classification. A total of 892,591 particles showing secondary features were selected, and a 3.35 Å model was generated using 154,013 particles after 2 rounds of ab initio and 1 round of NUR with C1 symmetry. Three million particles selected after removing non-protein junk particles were subjected to two rounds of HRC, using one good volume from NUR and seven decoy noise volumes. A total of 426,862 selected particles were refined to 2.82 Å by NUR with C3 symmetry. Polishing in Relion, HRC, and NUR was repeated twice. Finally, 404,729 particles were refined to 2.44 Å by NUR with C3 symmetry. Symmetry expansion and local classification identified ~10% of protomers in a different conformation. The particles from the minor class yielded a 2.93 Å map after local refinement and post-processing in Relion. Supplementary Figure 4 shows the image processing flowchart for the dataset. The image processing of EAAT3-X in 150 mM NMDG or 300 mM NaCl datasets was similar to the KCl dataset. The data processing information is in Supplementary Tables 2 and 3.

## Model building and refinement

Structures of hEAAT3g (PDB accession codes 6X2L, 6X3F, and 6X2Z) were first fitted into maps using ChimeraX[79]. The models were manually adjusted in COOT[80] and subjected to real-space refinement in Phenix[81]. The cross-validation was performed in Phenix by displacing atoms in the final model by 0.3 Å, refining the displaced model against the first unfiltered half-map (FSC-work). FSC curves were then calculated between the FSC-work model and the second unfiltered half-map (FSC-free) and between the refined model and the full map (FSC-sum). The structural figures were prepared in ChimeraX and Pymol (DeLano Scientific). The tunnel through the protein structure was calculated using CARVER3.0[82]. The pKa values of E374 were calculated using PROPKA 3.0[83]. The solvent-accessible surface areas of E374 were calculated using PISA[84].

## Proteoliposome reconstitution and solid-supported membrane (SSM) assay

The proteoliposome reconstitution and SSM assay were performed as previously described[21]. Liposomes were prepared using 1-palmitoyl-2-oleoyl-sn-glycero-3-phosphocholine (POPC, Avanti Polar Lipids), 1-palmitoyl-2-oleoyl-sn-glycero-3-phosphoethanolamine (POPE, Avanti Polar Lipids) and CHS at a 5:5:2 ratio. The lipids in chloroform were dried and rehydrated at 20 mg/ml by 10 freeze-thaw cycles in 50 mM HEPES-Tris buffer, pH 7.4, forming multilamellar liposomes. The liposomes were diluted to 4 mg/ml in a buffer containing 50 mM HEPES/NaOH, pH 7.4, 200 mM NaCl, 1 mM TCEP, and 1 mM L-aspartate and extruded 11 times through 400 nm polycarbonate membranes (Avanti Polar Lipids) using a syringe extruder (Avanti Polar Lipids). The unilamellar liposomes were destabilized by adding DDM to a 1:0.75 lipid-to-detergent ratio at 23 °C for 15 min. Purified MinCys EAAT3 and MinCys EAAT3 C343V were added to the liposomes and incubated for 30 min at 23 °C at a 1:10 protein-to-lipid ratio. To remove detergents, proteoliposomes were incubated with 100 mg/ml Bio-Beads SM-2 (Bio-Rad) for 1 h at 23 °C. Used beads were removed and replaced with new beads. The procedure was repeated 5 times, incubating for 1 h at 4 °C 3 times, overnight at 4 °C once, and 1 h at 4 °C once. The proteoliposomes were collected by centrifugation at 86,600g for 45 min at 4 °C. After removing the supernatant, the proteoliposomes were resuspended in a buffer containing 100 mM potassium phosphate, pH 7.4, and 2 mM MgSO$_4$ (SSM assay resting buffer). The resuspended proteoliposomes were subjected to freeze–thaw cycles in liquid nitrogen. The centrifugation and freeze–thaw steps were repeated 3 times for complete buffer exchange. The SSM assays were performed using a SURFE2R N1 instrument (Nanion Technologies). Briefly, proteoliposomes were extruded 11 times through 400 nm polycarbonate membranes and coated onto the SF-N1 sensor (Nanion Technologies) per the instrument manual. The nonactivating buffer containing 100 mM sodium phosphate, pH 7.4, and 2 mM MgSO$_4$ flowed through the sensor at a 200 µl/s flow rate to build the sodium and potassium gradients across the proteoliposome membranes. The transport-coupled

current was activated by flowing a buffer containing 100 mM sodium phosphate, pH 7.4, 2 mM $MgSO_4$, and 3 mM L-aspartate. Finally, the sensor was rinsed in the resting buffer. At least two sensors for each proteoliposome preparation were recorded, producing similar results.

## Reporting summary

Further information on research design is available in the Nature Portfolio Reporting Summary linked to this article.

## Data availability

The cryo-EM maps and atomic coordinates generated in this study have been deposited in the Electron Microscopy Data Bank (EMDB) and Protein Data Bank (PDB) under accession codes EMD-26985 (iOFS*-Glu, PDB entry 8CTC), EMD-26986 (OFS-Glu, PDB entry 8CTD), EMD-26997 (iOFS-K$^+$, PDB entry 8CUA), EMD-26998 (OFS-Apo$_{KCl}$, PDB entry 8CUD), EMD-27000 (iOFS-Apo, PDB entry 8CUI), EMD-27001 (OFS-Apo$_{NMDG}$, PDB entry 8CUJ), EMD-27006 (OFS-Na$^+$ bound, PDB entry 8CV2), and EMD-27007 (iOFS-Na$^+$ bound, PDB entry 8CV3). Previously published structural models and cryo-EM maps used in this research are available from PDB and EMDB, and the PDB accession codes 4P1A, 6UWL, 6X2Z, 6X2L, 6X3F, and EMDB accession codes EMD-22022, EMD-22023, EMD-22024. Source data are provided in this paper.

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

## Acknowledgements
We thank Drs. Xiaoyu Wang, Krishna Reddy, and Yun Huang for critical manuscript reading, data processing, and figure preparation. We thank Dr. Qianyi Wu for the help with the cellular uptake assays and Vishnu Ghani for the help with cell imaging. We thank Eric Robel for the suggestions on figures. We thank Eugene Chua at Simons EM Center at the New York Structural Biology Center (NYSBC) and Alice Paquette, Bing Wang, and William Rice at New York University Langone's cryo-EM laboratory for assistance with data collection. This work was supported by the National Institute of Neurological Disorders and Stroke grant R37NS085318 (O.B.). Some cryo-EM data collection was performed at the Simons Electron Microscopy Center and National Resource for Automated Molecular Microscopy located at the NYSBC, supported by grants from the Simons Foundation (SF349247), NYSTAR, and the NIH National Institute of General Medical Sciences (GM103310) with additional support from Agouron Institute (F00316), NIH (OD019994), and NIH (RR029300).

## Author contributions
B.Q. and O.B. conceived the projects; B.Q. performed all the experiments; B.Q. and O.B. analyzed the data and wrote the paper.

## Competing interests
The authors declare no competing interests.
