## [Peer Review File · Nature Communications]

Symport and antiport mechanisms of human glutamate transportersREVIEWER COMMENTS

Reviewer #1 (Remarks to the Author):

This a very detailed report on the structural intermediates of a neuronal glutamate transporter, which make the transporter competent for translocation across the membrane. Overall, several new cryo-EM structures are presented, which use crosslinking to stabilize an intermediate, iOFS*, which is present in several forms, depending on the ionic conditions in solution. In particular, a K⁺ bound state is identified, which confirms the location of the K⁺ binding site, previously determined by Rb⁺ binding to EAAT1, and computational studies. This structure also illustrates why K⁺ and Na⁺ binding to the Na₂ site are exclusive. In addition, conclusions are drawn about the protonation site of EAAT3. Obviously, the protonated form of the transporter cannot be resolved in cryo-EM, but pK_a value calculations using PROPKA reveal an anomalously high pK_a value of the E374 side chain, which had been implicated as a proton acceptor from mutagenesis studies. These calculations suggest that protonation of E374 facilitates Na⁺ binding and transition to the iOFS*, fully loaded state, which becomes translocation competent and leads to the inward-facing conformation. Overall, this paints a comprehensive structural picture of the substrate/Na⁺ cooperative binding mechanism, which is advancing the SLC1 transporter field, and, therefore is of significance.

Generally, the data appear to be of high quality and the methods and interpretations are sound, if maybe slightly over-interpreted with regard of the protonation site (PROPKA calculations are the only evidence). With that said, the quality of the manuscript could be further improved by the authors addressing the following points:

Major points:

- 1) In several previous studies, based on non-mammalian glutamate transporter homologs, it was observed that HP2 is actually closed in the apo conformation of the transporter. No stabilization of intermediates was used in those studies. However, in the present study, HP2 appears to be open in the iOFS apo state, and is only closed in the iOFS* apo state. Is there an explanation for this discrepancy?
- 2) The iOFS* is stabilized using crosslinking. Did the min-Cys EAAT3 construct still favor the inward facing conformation without crosslinking? Did the added Hg²⁺ have any effects in the transporter without the double-cys mutation? Was the Hg²⁺ observed in the cryo-EM maps?

Minor points:

Discussion on page 13, second paragraph. The conclusion that the proposed mechanism contradicts previously proposed mechanisms is not clear. This reviewer does not feel that previous literature proposed a wide-open HP2 as a translocation competent species. Maybe this paragraph needs further explanation.

Fig. 2d: It would help the reader to show the glutamate bound in the OFS-Glu cut through illustration, to be able to compare it to iOFD*-Glu in panel e.

Reviewer #2 (Remarks to the Author):

The authors performed extensive structural studies about human glutamate transporter EAAT3 and obtained 8 cryo-EM structures which represent different states (OFS and iOFS) during the substrate transport cycle. Structural analyses are solid, and the findings provide important new insights into mechanisms of ion coupling and substrate transport, which give a more complete illustration of the substrate transport process.

I only have some minor points:

1. As it is shown in Figure S2, both gel filtration profile and liposome-based transport activity have been examined for MinCys. Have you also tested these behaviors with purified MinCys K269C/W441C, which is used for crosslinking to capture the iOFS conformations? And what about the relative transport activity compared to WT EAAT3?

2. I suggest the authors to highlight the structures obtained from this study in Figure 6.

Reviewer #3 (Remarks to the Author):

Please see attached PDF file for "Comments for Author".

Qiu & Boudker report cryo-EM structures of the human EAAT3 transporter using a crosslinked mutant version, the EAAT3-X protein. Structures were found in the outward-facing state (OFS) and an occluded intermediate outward-facing state (iOFS), and in the presence or absence of potassium, sodium and/or glutamate. The structures were obtained from four different cryo-EM datasets resulting in eight different structures (in C1 and C3). In general, the paper is well written and illustrated, with space for improvement. Based on these structures the authors present a comprehensive ion coupling mechanism. In summary, a nice piece of work. However, the Reviewer has serious concerns on the used crosslinked human EAAT3-X protein and raises the pertinent question, if the obtained data/information reflect the wild-type form of EAAT3 or rather an artificial system: please see the detailed points below.

Major points

- **General:** Human EAAT3g was used as starting version to introduce the below mentioned numerous point mutations. According to Materials and Methods, hEAAT3g contains the mutations N178T and N195T.

1) Is this protein glycosylated at a different position?

2) If not (i.e., not glycosylated), how do the authors explain correct trafficking to the plasma membrane?

It would be helpful for the Readers to mention/discuss this issue briefly in the manuscript.

- **Suppl. Fig. S1, panel c:**

1) Two IFS structures are shown to represent the “dancing” motif rearrangement in response to sodium binding, but the cycle obviously begins in the OFS. Could the author comment on this and give an explanation why IFS structures are shown to explain an OFS rearrangement?

2) Please introduce distances in Å for all shown interaction (broken lines, e.g., between R447, and D444 and E374, etc.), so that the Reader and Reviewer can get an estimate on the strength of the corresponding interactions.

- **Suppl. Fig. 2:**

1) Please use different colors in panel b, not the same as in panel a: colors are already assigned to different constructs in panel a.

2) Panel b: Please also add a trace of the hEAAT3g for direct comparison together with MiniCys C343V 3 mM L-asp: hEAAT3g of course reconstituted under the same conditions as MiniCys and MiniCys C343V. Finally, please write “L-Asp” not “L-asp” (uncommon). This issue also found elsewhere in manuscript, e.g., Suppl. Table 2 (“L-glu” instead of “L-Glu”).

3) What is the peak at 17 ml, degradation or contamination? Please comment/provide data.

4) Please explain in the legend the SEC peak at about 19 ml. What hints from SDS-PAGE?

5) The peak at about 13 ml of MiniCys C343V could be an oligomer. Please comment on this possibility. Any experimental indication?

6) Please correct in legend: “MiniCys C343V” not “MinnCys C343V” (typo).

7) The statement that C343 mutations yielded denatured / inactive protein is made at least twice in the manuscript, but only C343V is shown. Please, either change to singular (mutant not mutants) referring to C343V only or provide experimental evidence for other C343 mutants, e.g., SSM assay traces.

- **Result:** Please revise sentence: “Sequence alignments showed that four of six EAAT3 cysteines are not conserved; the fifth is tryptophan in all other EAAT homologues”: the second part is confusing. Do you mean “...the fifth cysteine is replaced by a tryptophan in...”? Please use unambiguous wording.

- **Results:** After introduction of K269C and W441C into MinCys hEAAT3, I started asking myself how close this protein is to wild-type. Might we be looking at an artificial system not representing the wild-type anymore? These two mutations might have been described in the past in an archaeal homologue, but again was the model validated? Even if, we must agree that the MiniCys hEAAT3-X protein contains a significant amount of mutations (N178T, N195T, C9A, C100A, C158A, C219A, C256W, K269C, W441C) on top of being crosslinked with Hg²⁺ starting to be significantly different compared to wild-type hEAAT3.

As mentioned in my introductory part, this is a **major concern** of this Reviewer that needs to be addressed carefully.

- **Results:** Continuing on the previous point: Crosslinked EAAT3-X was found in the OFS and an iOFS conformations.

1) Since hEAAT3-X is fixed through crosslinking, how reliable is the observed iOFS?

2) Same question arises for previously published iOFS structures of archaeal homologues. I ask this because in contrast to the wild-type the mobility of the transporter in crosslinked hEAAT3-X **must be significantly reduced** (it's crosslinked). Thus, I would expect **increased flexibility** in the wild-type in contrast to crosslinked hEAAT3-X. Again, please provide also here argumentations to this important issue.

- **Results, sentence:** “Thus, crosslinking allows transport domain movements between OFS and iOFS*”. Again, it should be considered that the movement might be significantly reduced because of the crosslinking, not reflecting the wild-type. Clearly, **molecular dynamics simulation** might help here to prove that the crosslinked EAAT3-X reflects potentially the wild-type. Considering the concerns mentioned above **MD simulation are becoming essential** to corroborate the finding of the here presented work.

In summary, the Reviewer appreciates the structural work presented in this work, but has serious doubts on the relevance of the used crosslinked EAAT3-X model protein, e.g., does this model reflect the wild-type human EAAT3? Clearly, **MD simulation experiments** must be performed with the obtained structures in a lipid bilayer, of course removing the crosslink to warrant full mobility of the transporter (i.e., with K269 and W441). Additionally, the current structures with all wild-type residues (i.e., not MinCys) should be also considered for MD simulations. Such MD simulation might become crucial to corroborate the here presented structures of hEAAT3-X crosslinked by Hg²⁺.

Other points/issues

- **Please introduce page numbers:** in both, main manuscript and Suppl. Information

- **First sentence of Abstract:** Is the verb “to pump” appropriate for a transporter ?

- **Results, sentence:** “Together, these structures should describe the sequential binding of coupled solutes and visualize the potassium counter-transport.”

Please remove the word “transport”: “binding” is indeed visualized, but not the transport of potassium. A full simulation of EAAT3 containing all available information from this manuscript and Qiu et al. 2021, Sci. Adv. might perhaps “visualize the potassium counter-transport”. Currently, only snap-shots, no dynamics.

- **Sentence:** “...and the counter-transported potassium ions.”

The Reviewer suggests to use the singular, i.e., “...ion.” to avoid misunderstandings, i.e., that more than one potassium ions are transported per monomer and cycle.

- **Suppl. Fig. 4 and description in Materials and Methods:**

The description and presentation of the data processing is suboptimal and not fully clear. Concerning Image processing, 3rd section:

1) It is not entirely clear how “3 million particles ... after 2D cleaning” were selected, not in the description nor in the illustration. First, a 3.35 Å model was generated from remaining 154,013 particles. Did this model serve as input for a model-based particle picking procedure followed by 2D-classification, which ultimately led to this initial 3 million particles?

2) What is the origin of the 7 decoy noise *ab initio* volumes?

3) Symmetry expansion and local classification identified ~10% of protomers in a different conformation. Were these particles of obvious different conformation withdrawn from the final map?

4) Concerning Fig. 1, two volumes were calculated after symmetry expansion originating from, e.g., 14% (OFS-Glu) and 86% (iOFS*-Glu) of symmetry expanded particles. But according to the flowchart in Fig. S4 the “final” volume does not come from a symmetry expanded run and thus still contains the particles of the volume displaying the other conformation. Please comment.

5) When addressing the questions and rephrasing the corresponding paragraph, please provide more detailed information (Image processing, 3rd section) and redraw/correct and align Fig. S4 to the corrected text provided in Image processing, 3rd section.

- **Legend to Figure 1:** “HP2 in iOFS*-Glu, OFS-Glu, and iOFS- K⁺ fully occludes the ligand cavity; the cavities of iOFS-Apo and iOFS-Na⁺ remain solvent- accessible; HP2 of OFS-Apo in NMDG chloride and KCl and OFS-Na⁺ are wide-open.”

It would be helpful to color-code the information given in the caption concerning HP2 conformation. With eight different structures abbreviated (in fact in a logical) but complex way it would make sense to also provide a color-code to highlight certain aspects of the structures. Like this, the reader can more easily assess the information provided in the Figures.

The authors should alter the Figure accordingly using, e.g., a color highlighted states, which could in the text then be referred to as “(Fig. 1; states highlighted in green) throughout the manuscript.

- **Supplementary Tables 3 and 4:**

1) The entries in R.m.s. deviations; Bond lengths (Å), Bond angles (°) must have been mistyped and are certainly *vice versa* since an R.m.s. deviations entry of ~0.5 Å for bond lengths is not possible and a close to zero entry for R.m.s. deviations concerning bond angles very low.

Please check and address this issue.

2) To clarify which structure is which, the authors should provide the abbreviated state (e.g., OFS-Glu) of the protein in question in header of Tables S3 and S4.

- Figure 2 and L-Glu bound to iOFS-Glu:

1) It is not clear from the text nor from the Figure if L-Glu is also well visible in the OFS-Glu structure. If yes, is L-Glu binding the same in both structures?

2) Why was HP2 not colored in red in Fig. 2b? I suggest staying true to the color-code presented in Fig. S1a whenever possible.

3) What portion of the OFS-Glu and iOFS*-Glu were aligned for Fig. 2d and 2e?

Could the author please elaborate on this and redraw/correct Fig. 2 and/or adapt its caption?

- Figure 3. Potassium binding:

1) What is the rationale to contour the cryo-EM density at different sigma-values in Fig. 3a and 3b?

2) Some labels in Fig. 3a and 3b are hardly visible (T448, TM8, S331). Please improve color choice in function of background color. Check also the other Figures (also in Suppl. Information): sometimes also suboptimal.

3) In addition, please provide distances between the K⁺-atom and interaction partners. This will be helpful to Readers and Reviewers.

- Paragraph: Potassium binding seals the HP2 gate:

1) Sentence: "The ion is coordinated by carbonyl oxygen atoms of residues in the HP2 (408, 409, and 411) and HP1 (331) tips..."

What do the authors mean by "tips"? Did they mean the connecting loops between the α -helical segment of helical hairpins HP1 and HP2, respectively? Please define.

If so, is this truly a "tip"? At least the authors should define and label "tip" in Fig. S1a, provide the necessary information in the text, e.g., "...tip (see Supplementary Fig. 1a)", or come up with a different idea how to describe this region.

2) Sentence: "Here, we observe complete octahedral ion coordination with a mean distance between the ion and the coordinating oxygens of ~ 3.15 Å, within the expected range"

I would be interested in what an "expected range" is. Providing the distances between the K⁺-ion and interaction partners in Fig. 3a as suggested above, the authors could also provide a range in numbers here.

- Suppl. Fig. 7:

In panel Fig. S7, panel e M367 is not connected to the C α -trace of the structure suggesting that "smooth loops" in PyMol were activated, which in this context is misleading. The authors should correct the issue and redraw this panel.

- Suppl. Fig. 8:

As in Fig. 3, what is the rationale for the different contour-sigma-value to display cryo-EM density?

- Figure 6 and Molecular mechanism in general:

Sentence: "...transition into iOFS* and IFS." According to Fig. 6, it would be iOFS*-**Glu** and IFS-**Glu**. Since the abbreviations are in general already a bit confusing, sticking to them is of utmost importance.

Could the authors please provide the PDB-code for each known structure, implementing them either in Fig. 6 or the caption?

Sentence: "Therefore, iOFS has ~80 times higher affinity for glutamate than OFS."

Please provide the mathematical background for this statement and a minimal information for Readers.

- Please add, at least in Suppl. Information, a Figure/panel showing the trimeric hEAAT-X protein model – this structural view/information is currently missing (only density shown in Suppl. Fig. 4).

- What version of CryoSPARC was used for the calculations?

- **In Fig. S4:** CryoSPARC is written with "K" instead of a "C", please correct.

- Sometimes the authors use the abbreviation "NUC" instead of "NUR", please correct throughout the manuscript.

- For PyEM a reference is missing (Asarnow, D., Palovcak, E. & Cheng, Y. asarnow/pyem: UCSF pyem v0.5 (v0.5). *Zenodo*, doi:org/10.5281/zenodo.3576630 (2019), please cite it.

- **Sentence:** "To ensure symport and antiport of ions, the transport domain must undergo elevator transitions either when bound to glutamate/3Na⁺/proton or K⁺ ions and not when apo or bound to sodium only (**Fig. 1**)."

The mention of Fig. 1 in this sentence is unclear to the Reviewer: please comment.

- **Sentence:** "The cross-validation was performed by displacing atoms in the final model by 0.3 Å, refining the displaced model against the first unfiltered half- map (FSC-work). FSC curves were then calculated between the FSC-work model and the second unfiltered half-map (FSC-free) and between the refined model and the full map (FSC-sum)"

In the *Model building and refinement* section the cross-validation is explained. Could the authors please mention what type of software was used for these calculations?

- Why did purification buffers contain 1 mM L-Asp instead of 1 mM L-Glu?

Please also provide information in manuscript.

- The authors used QF R1.2/1.3 300 mesh gold grids (Quantifoil) to image crosslinked MiniCys EAAT3 K269C/W441C (EAAT3-X). Could the authors please specify what kind of gold grids they used, i.e., were they holey carbon coated gold grids or UltraAuFoil grids.

- In some cases, the space between number and unit is missing (e.g., 1mM instead of 1 mM). Please correct throughout the manuscript.

- **Sentence:** "Detergents were removed by incubation with 100 mg/ml Bio-Beads SM-2 (Bio-Rad) for 1h at 23°C, 1h at 4°C (3 times), overnight at 4°C, and 1h at 4°C."

Could the authors please clarify; fresh 100 mg/mL Bio-Beads were added after each round, meaning in total six batches? Currently unclear.

- **Sentence:** "The proteoliposomes were collected by centrifugation at 40,000 rpm for 45 min at 4°C"

Please either provide the rotor type or the centrifugation-speed in "g".

- **Sentence:** "The re-suspended was subjected to freeze-thaw in liquid nitrogen"

Isn't there a word missing?

Reviewer #1 (Remarks to the Author):

This a very detailed report on the structural intermediates of a neuronal glutamate transporter, which make the transporter competent for translocation across the membrane. Overall, several new cryo-EM structures are presented, which use crosslinking to stabilize an intermediate, iOFS*, which is present in several forms, depending on the ionic conditions in solution. In particular, a K⁺ bound state is identified, which confirms the location of the K⁺ binding site, previously determined by Rb⁺ binding to EAAT1, and computational studies. This structure also illustrates why K⁺ and Na⁺ binding to the Na₂ site are exclusive. In addition, conclusions are drawn about the protonation site of EAAT3. Obviously, the protonated form of the transporter cannot be resolved in cryo-EM, but pKa value calculations using PROPKA reveal an anomalously high pKa value of the E374 side chain, which had been implicated as a proton acceptor from mutagenesis studies. These calculations suggest that protonation of E374 facilitates Na⁺ binding and transition to the iOFS*, fully loaded state, which becomes translocation competent and leads to the inward-facing conformation. Overall, this paints a comprehensive structural picture of the substrate/Na⁺ cooperative binding mechanism, which is advancing the SLC1 transporter field, and, therefore is of significance.

Generally, the data appear to be of high quality and the methods and interpretations are sound, if maybe slightly over-interpreted with regard of the protonation site (PROPKA calculations are the only evidence).

We appreciate the reviewer's positive comments! We agree about the tentative nature of the conclusions on the protonation site and added a cautionary statement (lines 315-317).

1) In several previous studies, based on non-mammalian glutamate transporter homologs, it was observed that HP2 is actually closed in the apo conformation of the transporter. No stabilization of intermediates was used in those studies. However, in the present study, HP2 appears to be open in the iOFS apo state, and is only closed in the iOFS* apo state. Is there an explanation for this discrepancy?

If we understand correctly, the reviewer asks why HP2 is open in the apo OFS of EAAT3 but closed in the apo OFS of archaeal homologs GltPh and GltTk. We also found this an interesting question. The OFS-Apo of EAAT3 is structurally very similar to the sodium-bound OFS-Na, which may explain why it features an open HP2. We speculate that in EAAT3, the energetic balance between the apo conformation with closed HP2 and the conformation resembling OFS-Na with open HP2 is shifted toward the latter. Therefore, OFS-Apo resembles OFS-Na in EAAT3. In iOFS, additional interactions between HP2 and the scaffold favor a more closed hairpin conformation. Notably, HP2 of apo EAAT3 remains distorted and partially open even in the iOFS. We clarified this in the revised Discussion (lines 356-362).

2) The iOFS* is stabilized using crosslinking. Did the min-Cys EAAT3 construct still favor the inward facing conformation without crosslinking? Did the added Hg²⁺ have any effects in the transporter without the double-cys mutation? Was the Hg²⁺ observed in the cryo-EM maps?

We think that Hg-crosslinking is essential to maintain the transporter in OFS and iOFS states. Indeed, we recently collected Cryo-EM data on Hg-crosslinked min-Cys EAAT3 K269C/W441C frozen on grids immediately after adding L-cysteine. We found about 10% monomers in IFS, likely due to L-cysteine sequestered Hg²⁺ and breaking the crosslink. Therefore, we believe that min-Cys EAAT3 and uncross-linked min-Cys EAAT3 K269C/W441C favor inward-facing conformations.

We found some density between side chains of Cys269 and Cys441, which we assigned as Hg²⁺ ions (Supplementary Figure 5a and b). Based on these observations, we think that cross-linking is in place and necessary to keep the protein in OFS and iOFS.

We did not test the effects of Hg^{2+} on Min-Cys EAAT3. Hg^{2+} ions could bind to the remaining Cys343 of Min-Cys EAAT3, possibly inhibiting the transporter. However, in the cross-linked Min-Cys EAAT3 K269C/W441C, Cys343 is buried and has no adjacent density in Cryo-EM maps that could be attributed to Hg^{2+} . From the literature, HgCl_2 inhibited wild-type EAAT3 with the IC_{50} of 3.5 μM in cerebellar granule cells ¹. However, EAAT3 was unaffected by 2 μM cadmium in HeLa cells (Ref 41).

Minor points:

Discussion on page 13, second paragraph. The conclusion that the proposed mechanism contradicts previously proposed mechanisms is not clear. This reviewer does not feel that previous literature proposed a wide-open HP2 as a translocation competent species. Maybe this paragraph needs further explanation.

Thank you for the suggestion; we agree that the paragraph is ambiguous. Indeed, previous literature proposed that the translocation-incompetent conformations should have a wide-open HP2. In contrast, our structures show that HP2 is almost completely closed in translocation-incompetent iOFS-apo and iOFS- Na^+ conformations. We have revised the paragraph for clarity (lines 365-367).

Fig. 2d: It would help the reader to show the glutamate bound in the OFS-Glu cut through illustration, to be able to compare it to iOFD*-Glu in panel e.

We updated Fig. 2d as suggested.

Reviewer #2 (Remarks to the Author):

The authors performed extensive structural studies about human glutamate transporter EAAT3 and obtained 8 cryo-EM structures which represent different states (OFS and iOFS) during the substrate transport cycle. Structural analyses are solid, and the findings provide important new insights into mechanisms of ion coupling and substrate transport, which give a more complete illustration of the substrate transport process.

I only have some minor points:

1. As it is shown in Figure S2, both gel filtration profile and liposome-based transport activity have been examined for MinCys. Have you also tested these behaviors with purified MinCys K269C/W441C, which is used for crosslinking to capture the iOFS conformations? And what about the relative transport activity compared to WT EAAT3?

We appreciate the reviewer's positive comments. SSM assay is not suitable for comparing different constructs quantitatively because the signals can vary from sensor to sensor significantly. Instead, we measured uptake into cells expressing MinCys, MinCys K269C/W441C, and hEAAT3g, a glycosylation mutant showing activity similar to wild-type hEAAT3 ². MinCys EAAT3 shows similar activity to hEAAT3g, and the MinCys K269C/W441C has about 20 % activity, similar to the published data (Ref 41). We included these new results as new Supplementary Figures 2d and 2e and referred to in the Main text (lines 97-99).

2. I suggest the authors to highlight the structures obtained from this study in Figure 6.

Thank you for the suggestion. We highlighted the structures obtained from this study by labeling them in dark green in revised Figure 6. We also added the PDB code for known conformations of EAAT3 (labeled in blue).

Reviewer #3 (Remarks to the Author):

Qiu & Boudker report cryo-EM structures of the human EAAT3 transporter using a crosslinked mutant version, the EAAT3-X protein. Structures were found in the outward-facing state (OFS) and an occluded intermediate outward-facing state (iOFS), and in the presence or absence of potassium, sodium and/or glutamate. The structures were obtained from four different cryo-EM datasets resulting in eight different structures (in C1 and C3). In general, the paper is well written and illustrated, with space for improvement. Based on these structures the authors present a comprehensive ion coupling mechanism. In summary, a nice piece of work.

We thank the Reviewer for the positive outlook and the detailed comments on the manuscript. We appreciate the effort the Reviewer has invested in improving the paper.

However, the Reviewer has serious concerns on the used crosslinked human EAAT3-X protein and raises the pertinent question, if the obtained data/information reflect the wild-type form of EAAT3 or rather an artificial system: please see the detailed points below.

Major points

- **General:** Human EAAT3g was used as starting version to introduce the below mentioned numerous point mutations. According to Materials and Methods, hEAAT3g contains the mutations N178T and N195T. 1) Is this protein glycosylated at a different position?

EAAT3 has 3 predicted glycosylation sites: N43, N178, and N195. Mass spectrometry analysis of the wild-type protein showed the presence of peptides with no glycosylation on N43 (data not shown), suggesting the residue is not glycosylated. Consistently, we do not see extra density near N43 in our EM density maps.

If not (i.e., not glycosylated), how do the authors explain correct trafficking to the plasma membrane? It would be helpful for the Readers to mention/discuss this issue briefly in the manuscript.

Our published data show that hEAAT3g (EAAT3 N178T/N195T) and WT are functionally similar in oocytes, suggesting that they traffic similarly to the cell surface². Rather than glycosylation, the EAAT3 C-terminal tail has been implicated in proper trafficking^{3,4}.

- **Suppl. Fig. S1, panel c:** Two IFS structures are shown to represent the “dancing” motif rearrangement in response to sodium binding, but the cycle obviously begins in the OFS. Could the author comment on this and give an explanation why IFS structures are shown to explain an OFS rearrangement? 2) Please introduce distances in Å for all shown interaction (broken lines, e.g., between R447, and D444 and E374, etc.), so that the Reader and Reviewer can get an estimate on the strength of the corresponding interactions.

Thank you for the question. Before the current work, we determined the apo and sodium-bound conformations only in IFS. The rearrangements of the residues coordinating sodium and substrate are believed to be similar in OFS and IFS. So, we used IFS-Apo and IFS-Na⁺ to introduce the “dancing” motifs. We clarified this in the text (lines 45-46). We labeled the distances in the revised figure as suggested.

Suppl. Fig. 2: Please use different colors in panel b, not the same as in panel a: colors are already assigned to different constructs in panel a.

We changed the colors to be consistent throughout Supplementary Figure 2.

Panel b: Please also add a trace of the hEAAT3g for direct comparison together with MiniCys C343V 3 mM L-asp: hEAAT3g of course reconstituted under the same conditions as MiniCys and MiniCys C343V. Finally,

please write “L-Asp” not “L-asp” (uncommon). This issue also found elsewhere in manuscript, e.g., Suppl. Table 2 (“L-glu” instead of “L-Glu”).

Thank you for the suggestions. Different constructs cannot be quantitatively compared using SSM assay because of large variations between individual SSM sensors. The purpose of this panel is to show qualitatively that, in contrast to MinCys, MinCys C343V is inactive. We now include a comparison of activity in cell uptake assays for hEAAT3g, MinCys, and MinCys K269C/W441 in Supplementary Figure 2, panels d and e. We changed L-asp and L-glu to L-Asp and L-Glu, as requested.

What is the peak at 17 ml, degradation or contamination? Please comment/provide data. 4) Please explain in the legend the SEC peak at about 19 ml. What hints from SDS-PAGE?

Shown in the panel are preparative elution profiles of the purified proteins following protease cleavage of the GFP tag. The peaks at 17 and 19 ml are the PreScission protease and the cleaved Strep-GFP tag, respectively. We updated this information in the legend.

The peak at about 13 ml of MinCys C343V could be an oligomer. Please comment on this possibility. Any experimental indication?

The broad elution profile of the C343V mutant more likely suggests protein aggregation rather than the formation of specifically assembled oligomers. We included a corresponding comment in the revised figure legend. We did not further explore this mutant because, in our experience, such elution profiles indicate unstable denaturing protein.

Please correct in legend: “MinCys C343V” not “MinnCys C343V” (typo).

Fixed.

The statement that C343 mutations yielded denatured / inactive protein is made at least twice in the manuscript, but only C343V is shown. Please, either change to singular (mutant not mutants) referring to C343V only or provide experimental evidence for other C343 mutants, e.g., SSM assay traces.

We mutated C343 to alanine, glycine, serine, threonine, and valine and used fluorescence-detection SEC (FSEC) to screen the mutants. We now show the results in the revised Supplementary Figure 2a. Only MinCys, MinCys C343G, and MinCys C343V show FSEC behavior similar to hEAAT3g. We purified MinCys C343G, which has an SEC profile similar MinCys C343V, indicative of denatured protein (Supplementary Figure 2b).

Results: Please revise sentence: “Sequence alignments showed that four of six EAAT3 cysteines are not conserved; the fifth is tryptophan in all other EAAT homologues”: the second part is confusing. Do you mean “...the fifth cysteine is replaced by a tryptophan in...”? Please use unambiguous wording.

Fixed.

Results: After introduction of K269C and W441C into MinCys hEAAT3, I started asking myself how close this protein is to wild-type. Might we be looking at an artificial system not representing the wild-type anymore? These two mutations might have been described in the past in an archaeal homologue, but again was the model validated? Even if, we must agree that the MiniCys hEAAT3-X protein contains a significant amount of mutations (N178T, N195T, C9A, C100A, C158A, C219A, C256W, K269C, W441C) on top of being crosslinked with Hg²⁺ starting to be significantly different compared to wild-type hEAAT3. As mentioned in my introductory part, this is a **major concern** of this Reviewer that needs to be addressed carefully.

We thank the reviewer for this careful consideration. To address the reviewer’s concern, we measured the uptake of radioactive L-Asp into HEK293 cells expressing hEAAT3g, which we previously showed to be

functionally similar to the WT EAAT3, MinCys, and MinCys K269C/W441C (new Supplementary Figure 2d and e). MinCys has almost the same activity as hEAAT3g, and the MinCys K269C/W444C has about 20 % activity. Thus, while slower than the WT transporter, MinCys K269C/W444C still shows robust transport activity. We selected K269C/W444C mutant because it has been extensively validated by Kanner's and Grewers' groups as the double cysteine mutant, which, upon cross-linking, prevents the transporter from reaching IFS (Ref 41 and Ref 42).

- **Results:** Continuing on the previous point: Crosslinked EAAT3-X was found in the OFS and an iOFS conformations. 1) Since hEAAT3-X is fixed through crosslinking, how reliable is the observed iOFS?

2) Same question arises for previously published iOFS structures of archaeal homologues. I ask this because in contrast to the wild-type the mobility of the transporter in crosslinked hEAAT3-X **must be significantly reduced** (it's crosslinked). Thus, I would expect **increased flexibility** in the wild-type in contrast to crosslinked hEAAT3-X. Again, please provide also here argumentations to this important issue.

Thank you for these and the following thoughtful comments. We first note that iOFS occurs spontaneously in archaeal transporters reconstituted into lipid nanodiscs without crosslinking^{5,6}. The evolutionary conservation of iOFS from archaeal to human transporters points to its functional significance and is one of our unexpected findings.

EM imaging shows that crosslinking allows domain movements between OFS and iOFS. The state populations depend on the bound ligand, suggesting that the protein redistributes between these states in response to sodium, glutamate, or potassium binding. For example, the same crosslinked protein sample prepared in NMDG with and without the consequent addition of Na⁺ showed 93% in OFS and 81% in iOFS, respectively. This is not unexpected because Hg²⁺-mediated crosslinks allow for significant protein mobility, as we and others have observed^{6,7}.

We agree with the reviewer that the crosslink might affect the dynamics, i.e., the rates with which the transport domains move between OFS and iOFS. However, we are only concerned with discreet structures in this manuscript, not the dynamics. Crosslink might also skew the state populations favoring iOFS. If so, the crosslink will similarly alter the energy differences between iOFS and OFS regardless of the ligands. Therefore, the population changes upon binding of different ligands are meaningful, and we can extract relative binding affinities of the states from these population changes.

We have clarified these aspects in the revised Discussion.

- **Results, sentence:** "Thus, crosslinking allows transport domain movements between OFS and iOFS*". Again, it should be considered that the movement might be significantly reduced because of the crosslinking, not reflecting the wild-type. Clearly, **molecular dynamics simulation** might help here to prove that the crosslinked EAAT3-X reflects potentially the wild-type. Considering the concerns mentioned above **MD simulation are becoming essential** to corroborate the finding of the here presented work.

In summary, the Reviewer appreciates the structural work presented in this work, but has serious doubts on the relevance of the used crosslinked EAAT3-X model protein, e.g., does this model reflect the wild-type human EAAT3? Clearly, **MD simulation experiments** must be performed with the obtained structures in a lipid bilayer, of course removing the crosslink to warrant full mobility of the transporter (i.e., with K269 and W441). Additionally, the current structures with all wild-type residues (i.e., not MinCys) should be also considered for MD simulations. Such MD simulation might become crucial to corroborate the here presented structures of hEAAT3-X crosslinked by Hg²⁺.

We have carefully considered the suggestion to investigate the transport domain movements by MD simulations. However, such a study must be extensive to be meaningful because these are large domain movements difficult to recapitulate using MD. We think that the structures we visualize faithfully capture

the low-energy states of the transporter. We note that because we observe iOFS and OFS, it is likely that the entire conformational space connecting these states is accessible to the crosslinked transporter. Therefore, other states within the space must be comparatively high-energy intermediates because we do not detect them in the EM ensembles. It is also possible that there are states between iOFS and OFS. However, these hypothetical intermediates are beyond the scope of this paper. Finally, one might argue that the observed iOFS is different from the iOFS that would have occurred without the crosslinks. However, the iOFS we observe is nearly identical to the iOFS of archaeal transporters without crosslinks, suggesting that it is a conserved low-energy state.

Our main accomplishment in this paper is to provide structural snapshots depicting key events underlying ion coupling. While important, the details of the complete transporter energy landscape and the dynamic flexibility of the transport domain are beyond our scope.

Other points/issues

- **Please introduce page numbers:** in both, main manuscript and Suppl. Information - **First sentence of Abstract:** Is the verb “to pump” appropriate for a transporter ?

Thank you for the suggestion. We changed “pump” to “uptake” in the revised manuscript. (line 10)

- **Results, sentence:** “Together, these structures should describe the sequential binding of coupled solutes and visualize the potassium counter-transport. ” Please remove the word “transport”: “binding” is indeed visualized, but not the transport of potassium. A full simulation of EAAT3 containing all available information from this manuscript and Qiu et al. 2021, Sci. Adv. might perhaps “visualize the potassium counter- transport”. Currently, only snap-shots, no dynamics.

We changed “counter-transport” to “binding”. (line 104)

- **Sentence:** “...and the counter-transported potassium ions.” The Reviewer suggests to use the singular, i.e., “...ion.” to avoid misunderstandings, i.e., that more than one potassium ions are transported per monomer and cycle.

Done

- **Suppl. Fig. 4 and description in Materials and Methods:**

The description and presentation of the data processing is suboptimal and not fully clear. Concerning Image processing, 3rd section:

1) It is not entirely clear how “3 million particles ... after 2D cleaning” were selected, not in the description nor in the illustration. First, a 3.35 Å model was generated from remaining 154,013 particles. Did this model serve as input for a model-based particle picking procedure followed by 2D-classification, which ultimately led to this initial 3 million particles?

We rewrote the corresponding Methods section and provided additional details on data processing. The 3 million particles were selected after removing obvious non-protein junk (line 537).

2) What is the origin of the 7 decoy noise *ab initio* volumes?

We added: “The 7 decoy noise *ab initio* volumes were generated by running about 10 iterations of *ab initio* reconstruction using particles after 2D cleaning” at (lines 515-516) .

3) Symmetry expansion and local classification identified ~10% of protomers in a different conformation. Were these particles of obvious different conformation withdrawn from the final map?

See the response to the next comment.

4) Concerning Fig. 1, two volumes were calculated after symmetry expansion originating from, e.g., 14% (OFS-Glu) and 86% (iOFS*-Glu) of symmetry expanded particles. But according to the flowchart in Fig. S4 the “final” volume does not come from a symmetry expanded run and thus still contains the particles of the volume displaying the other conformation. Please comment.

We find ~7%-20% protomers in minor conformations after symmetry expansion and local 3D classification in the 4 datasets. As each protomer works independently, a protomer in the minor conformation comes from a trimer, in which the other two protomers are likely in the major conformation, as per its higher probability. It turns out that we obtain the highest-resolution maps of the major conformation when we align trimeric particles with the C3 symmetry imposed, keeping all trimers, including those that contain protomers in the minor conformation. The contributions from the minor conformation protomers are averaged out in the final maps. This approach produces higher-resolution maps than refining the major conformation protomers separately. This occurs likely because the larger size of the trimer leads to a more accurate alignment and 3D reconstructions. In contrast, we obtained the highest-resolution maps of the minor conformations when we aligned only the protomers in these states. Thus, we used these different strategies for the highest-resolution maps of the major and minor conformations. We rewrote the *Methods* section to make this clear.

5) When addressing the questions and rephrasing the corresponding paragraph, please provide more detailed information (Image processing, 3^d section) and redraw/correct and align Fig. S4 to the corrected text provided in Image processing, 3^d section.

We updated Fig. S4.

- **Legend to Figure 1:** “HP2 in iOFS*-Glu, OFS-Glu, and iOFS- K⁺ fully occludes the ligand cavity; the cavities of iOFS-Apo and iOFS-Na⁺ remain solvent- accessible; HP2 of OFS-Apo in NMDG chloride and KCl and OFS-Na⁺ are wide-open.” It would be helpful to color-code the information given in the caption concerning HP2 conformation. With eight different structures abbreviated (in fact in a logical) but complex way it would make sense to also provide a color-code to highlight certain aspects of the structures. Like this, the reader can more easily assess the information provided in the Figures.

The authors should alter the Figure accordingly using, e.g., a color highlighted states, which could in the text then be referred to as “(Fig. 1; states highlighted in green) throughout the manuscript.

We appreciate the reviewer’s suggestion. We update Figure 1, highlighting the states with different colors: green for “zipped”, pink for “unzipped”, and blue for wide-open HP2. We also colored the transport domains differently in every structure and maintained the color scheme throughout the manuscript for further clarity.

- **Supplementary Tables 3 and 4:**

1) The entries in R.m.s. deviations; Bond lengths (Å), Bond angles (°) must have been mistyped and are certainly *vice versa* since an R.m.s. deviations entry of ~0.5 Å for bond lengths is not possible and a close to zero entry for R.m.s. deviations concerning bond angles very low. Please check and address this issue.

Thank you for pointing it out. We corrected the mistake in the revised manuscript.

2) To clarify which structure is which, the authors should provide the abbreviated state (e.g., OFS-Glu) of the protein in question in header of Tables S3 and S4.

Done

- **Figure 2 and L-Glu bound to iOFS-Glu:**

1) It is not clear from the text nor from the Figure if L-Glu is also well visible in the OFS-Glu structure. If yes, is L-Glu binding the same in both structures?

We found density at the binding site in the OFS-Glu corresponding to the bound L-Glu. At the current resolution of 3.4 Å for the OFS-Glu, we do not see differences compared to iOFS*-Glu. We comment on this in the revised text (line 146).

2) Why was HP2 not colored in red in Fig. 2b? I suggest staying true to the color-code presented in Fig. S1a whenever possible.

Done.

3) What portion of the OFS-Glu and iOFS*-Glu were aligned for Fig. 2d and 2e? Could the author please elaborate on this and redraw/correct Fig. 2 and/or adapt its caption?

OFS-Glu and iOFS*-Glu in Fig. 2d and e were aligned on the scaffold domain. We updated the figure legend and Fig. 2d to show the bound L-Glu.

- Figure 3. Potassium binding:

1) What is the rationale to contour the cryo-EM density at different sigma-values in Fig. 3a and 3b?

2) Some labels in Fig. 3a and 3b are hardly visible (T448, TM8, S331). Please improve color choice in function of background color. Check also the other Figures (also in Suppl. Information): sometimes also suboptimal. 3) In addition, please provide distances between the K⁺-atom and interaction partners. This will be helpful to Readers and Reviewers.

Because the density of the bound K⁺ ion is very strong, we used a higher contour level for clarity. We revised Fig. 3a, making the contour level the same as in Fig. 3b. We updated the labeling and added distances.

- Paragraph: Potassium binding seals the HP2 gate:

1) Sentence: "The ion is coordinated by carbonyl oxygen atoms of residues in the HP2 (408, 409, and 411) and HP1 (331) tips..." What do the authors mean by "tips"? Did they mean the connecting loops between the α-helical segment of helical hairpins HP1 and HP2, respectively? Please define. If so, is this truly a "tip"? At least the authors should define and label "tip" in Fig. S1a, provide the necessary information in the text, e.g., "...tip (see Supplementary Fig. 1a)", or come up with a different idea how to describe this region.

The tips are the loops connecting the pairs of HP1 and HP2 helices. We labeled them in the revised Supplementary Fig. 1a.

2) Sentence: "Here, we observe complete octahedral ion coordination with a mean distance between the ion and the coordinating oxygens of ~3.15 Å, within the expected range". I would be interested in what an "expected range" is. Providing the distances between the K⁺-ion and interaction partners in Fig. 3a as suggested above, the authors could also provide a range in numbers here.

We included the distances in the revised Fig. 3a. The distance range of potassium and coordinating oxygen is 2.50-3.59 Å. The mean value reported in the literature is 2.84 Å⁸.

- Suppl. Fig. 7:

In panel Fig. S7, panel e M367 is not connected to the Cα-trace of the structure suggesting that "smooth loops" in PyMol were activated, which in this context is misleading. The authors should correct the issue and redraw this panel.

In supplementary Figures 7e and 7f, each panel shows M367 in two conformations – the one observed in the K⁺-bound transporter and the one observed in the L-Glu-bound transporter. In 7e, we show the protein structure in iOFS-K⁺ with M367 as sticks. Blue spheres show M367 from iOFS*-Glu superimposed on iOFS-K⁺, labeled “bound” M367. It is not connected to the main chain because it comes from a different structure. In 7f, the green spheres show M367 from iOFS-K⁺ superimposed onto iOFS*-Glu, labeled “apo” M367. We clarified these points in the revised legend.

- Suppl. Fig. 8:

As in Fig. 3, what is the rationale for the different contour-sigma-value to display cryo-EM density?

We made all contour levels the same.

- Figure 6 and Molecular mechanism in general:

Sentence: “...transition into iOFS* and IFS.” According to Fig. 6, it would be iOFS*-Glu and IFS- Glu. Since the abbreviations are in general already a bit confusing, sticking to them is of utmost importance. Could the authors please provide the PDB-code for each known structure, implementing them either in Fig. 6 or the caption?

We added the PDB codes for the published structures to Figure 6 and color-coded the published and new structures in blue and green, respectively.

Sentence: “Therefore, iOFS has ~80 times higher affinity for glutamate than OFS.” Please provide the mathematical background for this statement and a minimal information for Readers.

We added the relevant equation to the revised text (lines 373-374).

- Please add, at least in Suppl. Information, a Figure/panel showing the trimeric hEAAT-X protein model – this structural view/information is currently missing (only density shown in Suppl. Fig. 4).

We now show the models of iOFS-K⁺ OFS-Na⁺ in the revised Supplementary Fig. 4.

- What version of CryoSPARC was used for the calculations?

cryoSPARC v3.2.0, added (line 503)

- In Fig. S4: CryoSPARC is written with “K” instead of a “C”, please correct.

Fixed

- Sometimes the authors use the abbreviation “NUC” instead of “NUR”, please correct throughout the manuscript.

Fixed

- For PyEM a reference is missing (Asarnow, D., Palovcak, E. & Cheng, Y. asarnow/pyem: UCSF pyem v0.5 (v0.5). *Zenodo*, doi:org/10.5281/zenodo.3576630 (2019), please cite it.

Added the reference.

- **Sentence:** “To ensure symport and antiport of ions, the transport domain must undergo elevator transitions either when bound to glutamate/3Na⁺/proton or K⁺ ions and not when apo or bound to sodium only (Fig. 1).”

The mention of Fig. 1 in this sentence is unclear to the Reviewer: please comment.

For the successfully coupled symport of ions, the transport domain of EAAT3 should be able to undergo rapid elevator transitions only when fully bound to 3 Na⁺, one glutamate, and a proton, but not in partially

bound states. The successful antiport requires that only K⁺-bound, and not apo, transport domains undergo elevator transitions. We revised the text to refer specifically to translocation-competent and translocation-incompetent states in Figure 1. (lines 114-121)

- **Sentence:** "The cross-validation was performed by displacing atoms in the final model by 0.3 Å, refining the displaced model against the first unfiltered half- map (FSC-work). FSC curves were then calculated between the FSC-work model and the second unfiltered half-map (FSC- free) and between the refined model and the full map (FSC-sum)"

In the *Model building and refinement* section the cross-validation is explained. Could the authors please mention what type of software was used for these calculations?

Cross-validation was performed in Phoenix. We added it to the *Method*. (line 500)

- Why did purification buffers contain 1 mM L-Asp instead of 1 mM L-Glu? Please also provide information in manuscript.

We used L-Asp because it binds to the transporter tighter than L-Glu and provides a better means to stabilize the protein throughout the purification and crosslinking process. We added the explanation to the *Methods*. (lines 453-455)

- The authors used QF R1.2/1.3 300 mesh gold grids (Quantifoil) to image crosslinked MiniCys EAAT3 K269C/W441C (EAAT3-X). Could the authors please specify what kind of gold grids they used, i.e., were they holey carbon coated gold grids or UltraAuFoil grids.

The grids are holey carbon-coated gold grids. We added the info to the *Methods*. (line 494)

- In some cases, the space between number and unit is missing (e.g., 1mM instead of 1 mM). Please correct throughout the manuscript.

Fixed

- **Sentence:** "Detergents were removed by incubation with 100 mg/ml Bio-Beads SM-2 (Bio- Rad) for 1h at 23°C, 1h at 4°C (3 times), overnight at 4°C, and 1h at 4°C."

Could the authors please clarify; fresh 100 mg/mL Bio-Beads were added after each round, meaning in total six batches? Currently unclear.

Correct. We performed six incubations with fresh beads. We clarified this in the revised *Methods*. (lines 569-571)

- **Sentence:** "The proteoliposomes were collected by centrifugation at 40,000 rpm for 45 min at 4°C" Please either provide the rotor type or the centrifugation-speed in "g".

Changed to 86,660 g. (line 572)

- **Sentence:** "The re-suspended was subjected to freeze-thaw in liquid nitrogen" Isn't there a word missing?

Fixed. (line 575)

- 1 Fonfria, E., Vilaro, M. T., Babot, Z., Rodriguez-Farre, E. & Sunol, C. Mercury compounds disrupt neuronal glutamate transport in cultured mouse cerebellar granule cells. *J Neurosci Res* **79**, 545-553, doi:10.1002/jnr.20375 (2005).
- 2 Qiu, B., Matthies, D., Fortea, E., Yu, Z. & Boudker, O. Cryo-EM structures of excitatory amino acid transporter 3 visualize coupled substrate, sodium, and proton binding and transport. *Sci Adv* **7**, doi:10.1126/sciadv.abf5814 (2021).

- 3 Cheng, C., Glover, G., Banker, G. & Amara, S. G. A novel sorting motif in the glutamate transporter
excitatory amino acid transporter 3 directs its targeting in Madin-Darby canine kidney cells and
hippocampal neurons. *J Neurosci* **22**, 10643-10652 (2002).
- 4 Su, J. F. *et al.* Numb directs the subcellular localization of EAAT3 through binding the YxNxxF motif.
J Cell Sci **129**, 3104-3114, doi:10.1242/jcs.185496 (2016).
- 5 Huang, Y. *et al.* Use of paramagnetic (19)F NMR to monitor domain movement in a glutamate
transporter homolog. *Nat Chem Biol* **16**, 1006-1012, doi:10.1038/s41589-020-0561-6 (2020).
- 6 Wang, X. & Boudker, O. Large domain movements through the lipid bilayer mediate substrate
release and inhibition of glutamate transporters. *eLife* **9**, doi:10.7554/eLife.58417 (2020).
- 7 Chen, I. *et al.* Glutamate transporters have a chloride channel with two hydrophobic gates. *Nature*
591, 327-331, doi:10.1038/s41586-021-03240-9 (2021).
- 8 Harding, M. M. Metal-ligand geometry relevant to proteins and in proteins: sodium and potassium.
Acta crystallographica. Section D, Biological crystallography **58**, 872-874,
doi:10.1107/s0907444902003712 (2002).

REVIEWERS' COMMENTS

Reviewer #1 (Remarks to the Author):

This is a comprehensive revision of the original manuscript. I have no further concerns.

Reviewer #2 (Remarks to the Author):

The authors performed extensive structural studies about human glutamate transporter EAAT3 and obtained 8 cryo-EM structures which represent different states (OFS and iOFS) during the substrate transport cycle. In the new version, the authors have well addressed most of the comments and questions raised by reviewers. But it raises a bit of concerns about how well can MinCys K269C/W441C represent real transport states, as new assay data suggest that it has a dramatical decrease (~80%) in transport activity compared to wild type protein. Beside this point, this manuscript is in good quality and I suggest it to be published in Nature Communications.

Reviewer #3 (Remarks to the Author):

The authors have addressed satisfactorily the raised points/issues.

Thank you and congratulations

Reviewer #1 (Remarks to the Author):

This is a comprehensive revision of the original manuscript. I have no further concerns.

Thank you!

Reviewer #2 (Remarks to the Author):

The authors performed extensive structural studies about human glutamate transporter EAAT3 and obtained 8 cryo-EM structures which represent different states (OFS and iOFS) during the substrate transport cycle. In the new version, the authors have well addressed most of the comments and questions raised by reviewers. But it raises a bit of concerns about how well can MinCys K269C/W441C represent real transport states, as new assay data suggest that it has a dramatical decrease (~80%) in transport activity compared to wild type protein. Beside this point, this manuscript is in good quality and I suggest it to be published in Nature Communications.

Thank you for the comment and the support of our work. Our assay shows that the activity of MinCys K269/W441C is decreased, but the mutant is still active. We think that, most likely, the mutations affect the relative energies of the OFS and IFS as they are on the interface between the scaffold and transport domains in both states. These changes may affect the rates of transitions between the states and, therefore, the transport rates. However, it is unlikely that these mutations affect the conformations of the states, and thus the mutant should be suitable for structural studies. We also note that 20 % activity is often considered sufficient in functional studies of transporters. Assuming the wild-type transports with the rate of ca 10-100 glutamate molecules per second, our mutant transports 2-20 per second, still a very robust functional activity.

Reviewer #3 (Remarks to the Author):

The authors have addressed satisfactorily the raised points/issues.

Thank you and congratulations

Thank you, we appreciate the effort all reviewers put into improving the manuscript.